# Insights into herpesvirus assembly from the structure of the pUL7:pUL51 complex

Benjamin G Butt[1], Danielle J Owen[1], Cy M Jeffries[2], Lyudmila Ivanova[1], Chris H Hill[1], Jack W Houghton[3], Md Firoz Ahmed[1], Robin Antrobus[3], Dmitri I Svergun[2], John J Welch[4], Colin M Crump[1], Stephen C Graham[1]*

[1]Department of Pathology, University of Cambridge, Cambridge, United Kingdom; [2]European Molecular Biology Laboratory (EMBL) Hamburg Site, Hamburg, Germany; [3]Cambridge Institute for Medical Research, University of Cambridge, Cambridge, United Kingdom; [4]Department of Genetics, University of Cambridge, Cambridge, United Kingdom

**Abstract** Herpesviruses acquire their membrane envelopes in the cytoplasm of infected cells via a molecular mechanism that remains unclear. Herpes simplex virus (HSV)−1 proteins pUL7 and pUL51 form a complex required for efficient virus envelopment. We show that interaction between homologues of pUL7 and pUL51 is conserved across human herpesviruses, as is their association with *trans*-Golgi membranes. We characterized the HSV-1 pUL7:pUL51 complex by solution scattering and chemical crosslinking, revealing a 1:2 complex that can form higher-order oligomers in solution, and we solved the crystal structure of the core pUL7:pUL51 heterodimer. While pUL7 adopts a previously-unseen compact fold, the helix-turn-helix conformation of pUL51 resembles the cellular endosomal complex required for transport (ESCRT)-III component CHMP4B and pUL51 forms ESCRT-III–like filaments, suggesting a direct role for pUL51 in promoting membrane scission during virus assembly. Our results provide a structural framework for understanding the role of the conserved pUL7:pUL51 complex in herpesvirus assembly.

**\*For correspondence:**
scg34@cam.ac.uk

**Competing interests:** The authors declare that no competing interests exist.

## Introduction

Herpesviruses are highly prevalent human and animal pathogens that cause life-long infections and result in diseases ranging from cold sores and genital lesions (herpes simplex virus, HSV) to viral encephalitis (HSV-1), congenital birth defects (human cytomegalovirus, HCMV) and cancer (e.g. Kaposi's sarcoma associated herpesvirus, KSHV) (*Evans et al., 2013*; *Virgin et al., 2009*). Herpesviruses share conserved virion morphology, their DNA genome-containing capsids being linked to glycoprotein-studded limiting membranes via a proteinaceous layer called *tegument*, and a conserved assembly pathway whereby final envelopment of the DNA-containing capsids occurs in the cytoplasm (reviewed in *Mettenleiter et al., 2009*; *Owen et al., 2015*). While herpesviruses are known to extensively remodel the intracellular architecture of infected cells (*Das et al., 2007*), the molecular mechanisms by which they direct intracellular membranes to envelop nascent virions remain unclear.

HSV-1 tegument proteins pUL7 and pUL51 promote virus assembly by stimulating the cytoplasmic wrapping of nascent virions (*Albecka et al., 2017*; *Roller and Fetters, 2015*). pUL7 and pUL51 form a complex that co-localizes with Golgi markers both during infection and when co-transfected into cells (*Albecka et al., 2017*; *Roller and Fetters, 2015*; *Nozawa et al., 2003*), palmitoylation of residue Cys9 being required for pUL51 membrane association (*Nozawa et al., 2003*). Deletion of pUL7, pUL51, or both proteins from HSV-1 causes a 5- to 100-fold decrease in virus replication (*Albecka et al., 2017*; *Nozawa et al., 2005*; *Roller et al., 2014*) and cells infected with HSV-1 lacking pUL7 and pUL51 accumulate unenveloped capsids in the cytoplasm (*Albecka et al., 2017*). Similar results have been observed in other α-herpesviruses. pORF53 and pORF7, the pUL7 and

**eLife digest** Most people suffer from occasional cold sores, which are caused by the herpes simplex virus. This virus causes infections that last your entire life, but for the most part it lies dormant in your cells and reactivates only at times of stress. When it reactivates, the virus manipulates host cells to make new virus particles that may spread the infection to other people. Like many other viruses, herpes simplex viruses also steal jelly-like structures known as membranes from their host cells to form protective coats around new virus particles.

In cells from humans and other animals, proteins belonging to a molecular machine known as ESCRT form filaments that bend and break membranes as the cells require. Many viruses hijack the ESCRT machinery to wrap membranes around new virus particles. However, herpes simplex viruses do not follow the usual rules for activating this machine. Instead, they rely on two viral proteins called pUL7 and pUL51 to hot-wire the ESCRT machinery. Previous studies have shown that these two proteins bind to each other, but it remained unclear how they work.

Butt et al. used a combination of biochemical and biophysical techniques to solve the three-dimensional structures of pUL7 and pUL51 when bound to each other. The experiments determined that the structure of pUL51 resembles the structures of different components in the ESCRT machinery. Like the ESCRT proteins, pUL51 formed filaments, suggesting that pUL51 bends membranes in cells and that pUL7 blocks it from doing so until the time is right. Further experiments showed that the equivalents of pUL7 and pUL51 in other members of the herpes virus family also bind to each other in a similar way.

These findings reveal that herpes simplex viruses and their close relatives have evolved a different strategy than many other viruses to steal membranes from host cells. Interfering with this mechanism may provide new avenues for designing drugs or improving vaccines against these viruses. The pUL7 and pUL51 proteins may also inspire new tools in biotechnology that could precisely control the shapes of biological membranes.

pUL51 homologues from varicella-zoster virus (VZV), co-localize with *trans*-Golgi markers in infected cells (*Selariu et al., 2012*; *Wang et al., 2017*) and deletion of pORF7 causes a defect in cytoplasmic envelopment (*Jiang et al., 2017*). Similarly, deletion of pUL7 or pUL51 from pseudorabies virus (PrV) causes defects in virus replication and the accumulation of cytoplasmic unenveloped virions (*Klupp et al., 2005*; *Fuchs et al., 2005*), and PrV pUL51 co-localizes with Golgi membranes during infection (*Klupp et al., 2005*).

Homologues of pUL7 and pUL51 can be identified in β- and γ-herpesviruses, although pUL51 homologues lack significant sequence similarity with α-herpesvirus pUL51 and their homology is inferred from their conserved positions in virus genomes (*Lenk et al., 1997*; *Campadelli-Fiume et al., 2007*). The putative pUL51 homologue pUL71 from HCMV, a β-herpesvirus, associates with the Golgi compartment when expressed in isolation and with Golgi-derived virus assembly compartments during infection (*Dietz et al., 2018*). Deletion of pUL71 causes defects in HCMV replication, characterized by aberrant virus assembly compartments (*Womack and Shenk, 2010*) and defects in secondary envelopment (*Schauflinger et al., 2011*). Similarly, the HCMV pUL7 homologue pUL103 co-localizes with Golgi markers when expressed alone or during infection, and deletion of pUL103 causes a loss of assembly compartments, reductions in virus assembly and defects in secondary envelopment (*Ahlqvist and Mocarski, 2011*). Relatively little is known about the pUL7 and pUL51 homologues from γ-herpesviruses. Both the pUL7 and pUL51 homologues from murine γ-herpesvirus 68 are essential for virus replication (*Song et al., 2005*). The putative pUL51 homologue BSRF1 from Epstein-Barr virus associates with Golgi membranes and siRNA knock-down of BSRF1 in B95-8 cells prevents virion production (*Yanagi et al., 2019*). The KSHV homologue of pUL7, pORF42, is similarly required for efficient virion production (*Butnaru and Gaglia, 2019*). While a direct interaction has not been shown for the pUL7 and pUL51 homologues from β- or γ-herpesviruses, the EBV homologues BBRF2 and BSRF1 have been shown to co-precipitate from transfected cells (*Yanagi et al., 2019*).

Definitive molecular characterization of pUL7 and pUL51 function in HSV-1 or other herpesviruses has been hampered by their lack of homology to any proteins of known structure or function.

However, a recent study of HCMV hypothesized that the pUL51 homologue pUL71 may act as a viral endosomal sorting complex required for transport (ESCRT)-III component (*Streck et al., 2018*). We characterized the pUL7:pUL51 complex by solution scattering and solved the atomic-resolution structure of the proteolysis-resistant core of this complex using X-ray crystallography. pUL7 comprises a single globular domain that binds one molecule of pUL51 via a hydrophobic surface, a second molecule of pUL51 being recruited to the solution complex via the N-terminal region of pUL51. While the fold of pUL7 is not similar to any known structure, the α-helical pUL51 protein shares unanticipated structural similarity to components of the ESCRT-III membrane-remodeling machinery. Like cellular ESCRT-III component CHMP4B, pUL51 is capable of forming long filaments. Furthermore, we show that formation of the pUL7:pUL51 complex and its association with the *trans*-Golgi network is conserved across α-, β- and γ-herpesviruses, consistent with a conserved function for this complex in herpesvirus assembly.

## Results

### HSV-1 pUL7 and pUL51 form a 1:2 heterotrimer in solution

Full-length HSV-1 pUL7 and pUL51 were co-expressed in *Escherichia coli*, the palmitoylation site of pUL51 (Cys9) having been mutated to serine to avoid aberrant disulfide bond formation (*Figure 1—figure supplement 1*). Proteins were co-expressed and co-purified because pUL51 (25.5 kDa) formed large soluble aggregates when purified alone (*Figure 1—figure supplement 1*) and pUL7 (33.0 kDa) was extremely prone to aggregation upon removal of the GST purification tag when purified in the absence of pUL51. Multi-angle light scattering (MALS) analysis showed the complex to elute from size-exclusion chromatography (SEC) as two peaks with molecular masses of 79.0 ± 1.8 kDa and 165.5 ± 1.1 kDa (*Figure 1A*), consistent with pUL7 and pUL51 forming a 1:2 heterotrimer in solution (calculated mass from amino acid sequence 84.5 kDa) that dimerizes at higher concentrations to form a 2:4 heterohexamer (calculated mass 169 kDa). However, pUL51 of the co-purified complex was prone to degradation, frustrating crystallization attempts (*Figure 1A*). Prior sequence analysis (*Oda et al., 2016*; *Nozawa et al., 2003*) and our bioinformatics (*Figure 1—figure supplement 2*) suggested that the C-terminal region of pUL51 lacks regular secondary structural elements and is disordered. Consistent with this prediction, SEC with inline small-angle X-ray scattering (SAXS) showed the pUL7:pUL51 complex to be extended. The 1:2 and 2:4 complexes have radii of gyration ($R_g$) of 4.3 and 4.8 nm, with maximum particle dimensions ($D_{max}$) of ~18 nm and 20 nm, respectively (*Figure 1B, J, K* and *Supplementary file 1*–Table S1). *Ab initio* shape analysis was performed by fitting the 2:4 scattering curve to a dummy-atom model, or simultaneously fitting both scattering curves to a dummy-residue model, with the imposition of P2 symmetry. The models thus obtained are consistent with the pUL7:pUL51 complex comprising a folded core with an extended region of poorly-ordered amino acids (*Figure 1C and D*). In agreement with this, dimensionless Kratky plots of the 1:2 and 2:4 complex SAXS data shows both to have maxima above $sR_g = \sqrt{3}$ (*Figure 1L*) with extended tails observed in the corresponding probable frequency of real-space distances (p(r) profiles) at longer vector-length distances (*Figure 1K*).

Previous truncation analysis had shown residues 29–170 of pUL51 to be sufficient for pUL7 binding (*Albecka et al., 2017*). However, neither pUL7 in complex with pUL51 residues 29–170, nor with pUL51 residues 1–170, proved amenable to crystallization. Mass spectrometry analysis identified a smaller protein species, evident whenever the pUL7:pUL51(1–170) was analyzed by SDS-PAGE, as pUL51 residues 8–142. On the assumption that this represented the proteolysis-resistant fragment of pUL51, pUL7 was co-expressed and co-purified with pUL51(8–142). This protein complex could be readily purified and was monodisperse in solution, SEC-MALS showing the pUL7:pUL51(8–142) complex to have a mass of 61.5 ± 3.1 kDa, consistent with a 1:2 complex (calculated mass 63.1 kDa) as observed for full-length pUL7:pUL51 (*Figure 1E*). SEC-SAXS analysis (*Figure 1G*) showed the pUL7:pUL51(8–142) complex to be much more compact ($R_g$ = 3.0 nm; $D_{max}$ = 11.5 nm; *Figure 1K*; *Supplementary file 1*–Table S1). The Gaussian-like appearances of a dimensionless Kratky plot of the pUL7:pUL51(8–142) scattering data, which is centered on $sR_g$ of $\sqrt{3}$ (*Figure 1L*), and of the corresponding p(r) profile (*Figure 1K*) are consistent with the protein having a globular fold. *Ab initio* shape analysis of this data reveals that the pUL7:pUL51(8–142) complex visually resembles the folded core of the full-length complex (*Figure 1H and I*).

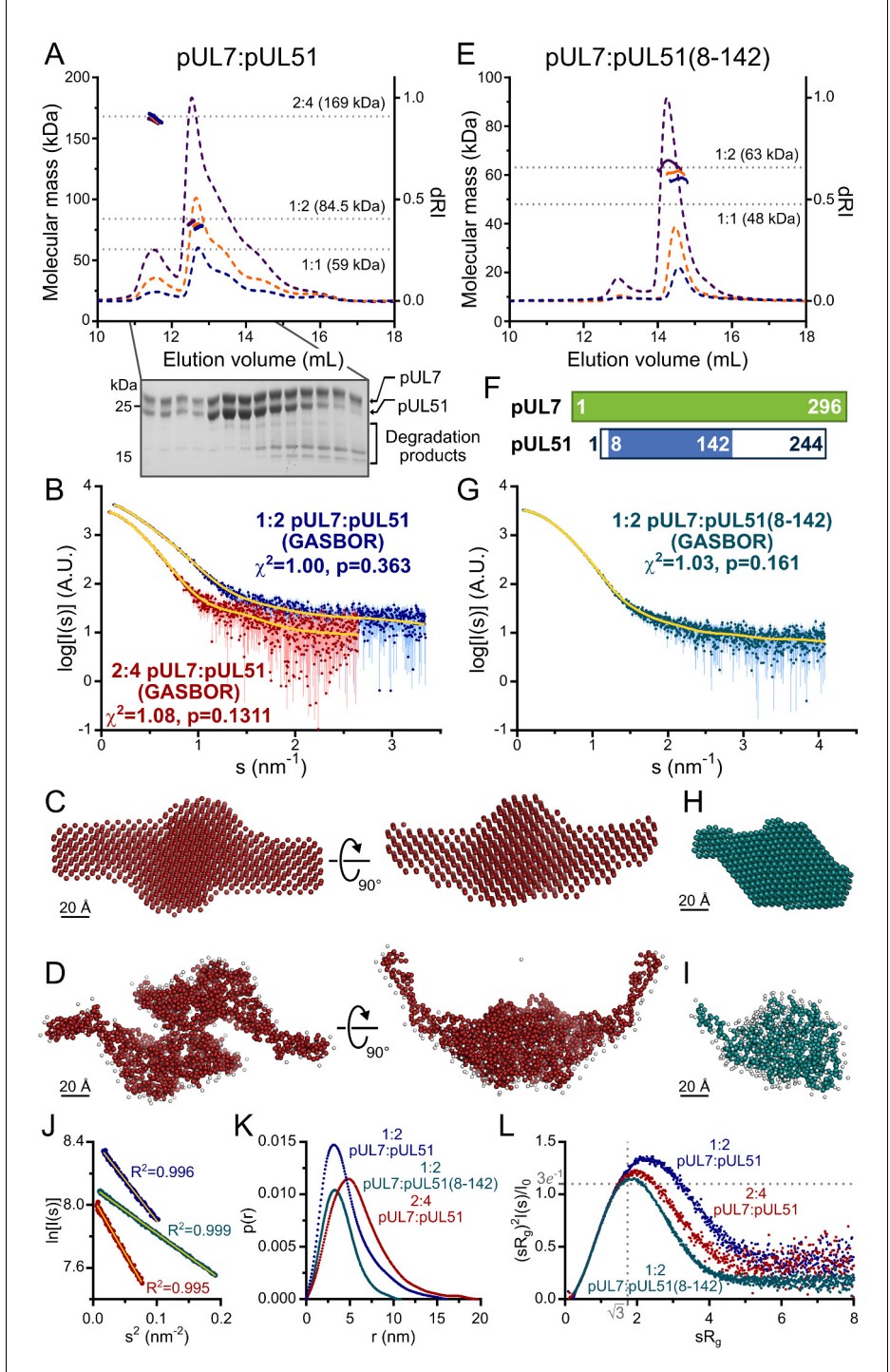

**Figure 1.** HSV-1 pUL7:pUL51 forms a 1:2 heterotrimeric complex in solution. (**A**) SEC-MALS analysis of recombinant full-length pUL7:pUL51 complex. Weight-averaged molecular masses (colored solid lines) are shown across the elution profiles (normalized differential refractive index, dRI, colored dashed lines) for samples injected at 2.4, 4.9 and 9.7 mg/mL (blue, orange and purple, respectively). The expected molecular masses for 1:1, 1:2 and 2:4 pUL7:pUL51 complexes are shown as dotted horizontal lines. (**B**) Averaged SAXS profiles through SEC elution peaks corresponding to 1:2 (blue) and 2:4 (red) complexes of pUL7:pUL51. Fits of representative GASBOR *ab initio* dummy-residue models to the scattering curves for each complex are shown in yellow. $\chi^2$, fit quality. p, Correlation Map (CorMap) *P*-value of systematic deviations between the model fit and scattering data (61). (**C**) Refined DAMMIN dummy-atom model reconstruction of the 2:4 pUL7:pUL51 complex, shown in two orthogonal

*Figure 1 continued on next page*

*Figure 1 continued*

orientations. (D) Representative GASBOR dummy-residue model of the 2:4 pUL7:pUL51 complex, shown in two orthogonal orientations. This model comprises an anti-parallel dimer of heterotrimers, although we note that parallel dimers are also consistent with the scattering data. (E) SEC-MALS of pUL7:pUL51(8-142) complex. Elution profiles and molecular masses are shown as in (A) for recombinant pUL7:pUL51(8–142) injected at 0.6, 1.1 and 3.9 mg/mL (blue, orange and purple, respectively). (F) Schematic representation of pUL7 and pUL51. (G) Averaged SEC-SAXS profile through pUL7:pUL51(8–142) elution peak. Fit of a representative GASBOR *ab initio* dummy-residue model to the scattering curve is shown in yellow. (H) Refined DAMMIN dummy-atom model reconstruction of pUL7:pUL51(8–142) complex. (I) Representative GASBOR dummy-residue model of pUL7:pUL51(8-142). (J) Plot of the Guinier region ($sR_g$ < 1.3) for SAXS profiles shown in (B) and (G). The fit to the Guinier equation (yellow) is linear for each curve, as expected for aggregate-free systems. (K) p(r) vs r profiles for SAXS profiles shown in (B) and (G). (L) Dimensionless Kratky plot of SAXS profiles shown in (B) and (G). The expected maximum of the plot for a compact globular domain that conforms to the Guinier approximation is shown ($sR_g = \sqrt{3}$, $(sR_g)^2 I(s)/I_0 = 3e^{-1}$, grey dotted lines).

The online version of this article includes the following figure supplement(s) for figure 1:

**Figure supplement 1.** HSV-1 pUL51 forms large soluble aggregates when purified in isolation.

**Figure supplement 2.** Predicted secondary structure of pUL7 and pUL51 homologues from representative human α-, β- and γ-herpesviruses.

## Structure of pUL7 in complex with pUL51(8–142)

The pUL7:pUL51(8–142) complex was crystallized and its structure was solved by four-wavelength anomalous dispersion analysis of a mercury acetate derivative. The structure of native pUL7:pUL51 (8–142) was refined to 1.83 Å resolution with residuals $R$ = 0.195, $R_{free}$ = 0.220 and excellent stereo-chemistry, 99% of residues occupying the most favored region of the Ramachandran plot (*Supplementary file 1*–Table S2). The asymmetric unit contained four copies of pUL7 residues 11–234 and 253–296 plus eight residues from the C-terminal purification tag (see below) and four copies of pUL51 residues 24–89 and 96–125, the remaining residues of pUL7 and pUL51(8–142) being absent from electron density and presumably disordered.

Strikingly, the molecules of pUL7 and pUL51 in the structure were arranged as a hetero-octamer, with single β-strands from each pUL7 and pUL51 molecule in the asymmetric unit forming a central β-barrel (*Figure 2A*). Closer inspection revealed that the pUL7 strands in this β-barrel comprised the C-terminal amino acids encoded by the restriction site and from the human rhinovirus 3C protease recognition sequence that remained following proteolytic removal of the GST purification tag. We therefore suspected that this hetero-octameric pUL7:pUL51 arrangement was an artefact of crystallization. SEC-MALS of a pUL7:pUL51(8–142) construct where the purification tag was moved to the N terminus of pUL7, and would thus be unlikely to form the same β-barrel observed in the crystal structure, yielded the same 1:2 pUL7:pUL51 heterotrimeric stoichiometry as observed with C-terminally tagged pUL7 (*Figure 2—figure supplement 1A*). Removal of residues 8–40 from pUL51, including residues 24–40 that form part of the β-barrel, yielded a 1:1 heterodimeric complex of pUL7 and pUL51(41–142) as determined by SEC-MALS (*Figure 2—figure supplement 1B*), although we note that this truncated complex had reduced solubility. Taken together, these results suggest that pUL7 and pUL51 residues 41–142 assemble to form a heterodimeric 'core' complex and that recruitment of the additional pUL51 molecule in the native heterotrimeric complex is mediated by the N-terminal region (residues 8–40) of pUL51.

The core heterodimeric complex formed by pUL7 residues 11–296 and pUL51 residues 41–125 is shown in *Figure 2B*. pUL7 comprises two short N-terminal α-helices followed by a compact globular fold with a mixed α-helical and β-sheet topology containing a central anti-parallel β-sheet and two mostly-buried α-helices that are surrounded by a β-hairpin and additional helices (*Figure 2—figure supplement 2*). Structure-based searches of the Protein Data Bank did not reveal any other domains with a similar fold, which we will henceforth refer to as the Conserved U$_L$7(Seven) Tegument Assembly/Release Domain (CUSTARD) fold. pUL51 comprises a hydrophobic loop region followed by a helix-turn-helix. The interaction with pUL7 is extensive and largely hydrophobic in nature (*Figure 2*): The hydrophobic loop of pUL51 (residues 45–50, sequence LLPAPI) interacts with pUL7 helix α2 and with a hydrophobic pocket formed by sheets β1 and β6, helices α4 and α7 and the C-terminal tail of pUL7; hydrophobic residues of pUL51 helix α1 interact with a hydrophobic face of pUL7 helix α8;

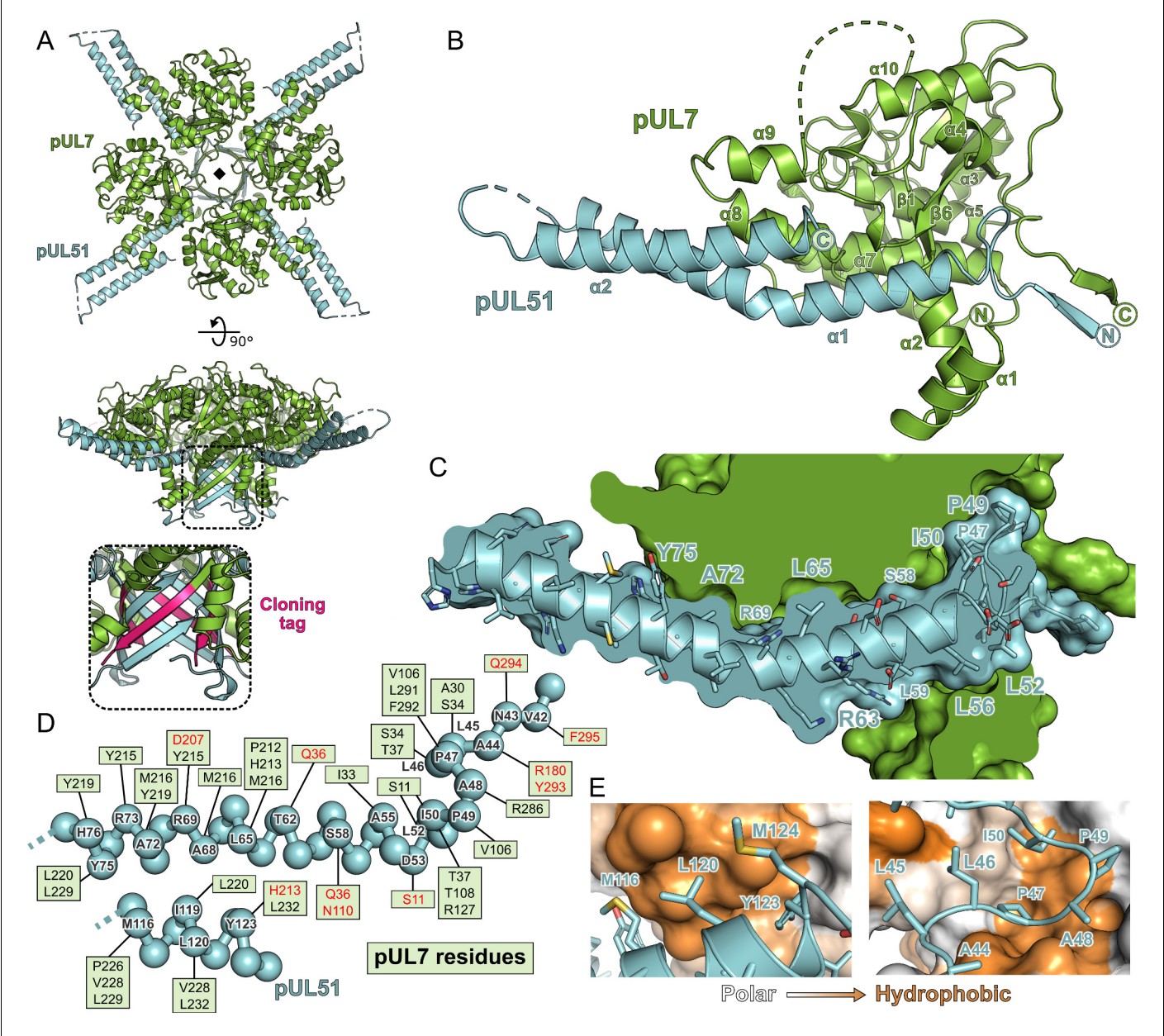

**Figure 2.** Structure of pUL7 in complex with pUL51. (A) Hetero-octamer of pUL7 and pUL51(8–142) observed in the crystallographic asymmetric unit. pUL7 and pUL51 are shown as green and cyan ribbons, respectively, in two orthogonal orientations. Inset shows residues arising from the pUL7 cloning tag (pink) that form an eight-stranded β-barrel with residues from pUL51. (B) Core heterodimer of pUL7 (residues 11–296) and pUL51 (residues 41–125). Selected secondary structure elements are labelled. (C) 'Cut-through' molecular surface representation of pUL7 (green) showing the intimate interaction interface with the hydrophobic loop and helix α1 of pUL51 (cyan). pUL51 side chains are shown as sticks. (D) Molecular interactions between pUL51 (cyan) and pUL7 (boxed residue names). Hydrophobic and hydrogen bond interactions are in black and red typeface, respectively. (E) Molecular surface representation of pUL7, colored by residue hydrophobicity from *white* (polar) to *orange* (hydrophobic). pUL51 is represented as a cyan ribbon with selected side chains shown.

The online version of this article includes the following figure supplement(s) for figure 2:

**Figure supplement 1.** SEC-MALS of truncated pUL7:pUL51 complexes.

**Figure supplement 2.** The CUSTARD fold of pUL7.

**Figure supplement 3.** Cross-linking mass spectrometry analysis and pseudo-atomic modelling of the pUL7:pUL51(8–142) solution heterotrimer.

and hydrophobic residues from the C-terminal portion of pUL51 helix α2 interact with hydrophobic residues from pUL7 helices α8 and α9 (*Figure 2C–E*).

Chemical cross-linking and mass spectrometry was used to further characterize the interaction between pUL7 and pUL51 in solution. As shown in *Figure 2—figure supplement 3A*, incubation of the pUL7:pUL51(8–142) complex with either disuccinimidyl sulfoxide (DSSO) or disuccinimidyl dibutyric urea (DSBU) yielded species with masses corresponding to 1:1 or 1:2 pUL7:pUL51 complexes, plus some higher-order species. Analysis of these cross-linked complexes by MS3 mass spectrometry identified multiple cross-links between pUL7 and pUL51 residues (*Supplementary file 1*–Table S3). Five of these crosslinks were not compatible with the heterodimer crystal structure, suggesting that they were formed by the other molecule of pUL51 in the heterotrimer, whereas other cross-links could have been formed by either pUL51 molecule. Multiple pseudo-atomic models of the 1:2 pUL7: pUL51(8–142) solution heterotrimer were thus generated by fitting the SAXS profile using the core heterodimer structure, a second copy of pUL51 residues 41–125, and permutations of the feasible chemical cross-linking restraints. Of the 80 models thus generated, half could not simultaneously satisfy all crosslinking restraints and were discarded. The other models all fit the pUL7:pUL51(8–142) SAXS profile well ($\chi^2$ < 1.26). These models showed the second copy of pUL51 to have the same general orientation relative to pUL7, binding near pUL7 helices α1, α2, α6, α7, and the loop between helices α7 and α8 (*Figure 2—figure supplement 3C*). However, the precise orientations of this second pUL51 copy differed, as did the locations of the pUL51 termini. The observed variability is within the resolution limits provided by SAXS, although it may also point to co-existence of alternate conformations, i.e. that the second copy of pUL51 does not adopt one well-defined conformation in solution.

## The interaction between pUL7 and pUL51 is conserved across herpesviruses, but the molecular details of the interface have diverged

The α-, β- and γ-herpesvirus families diverged approximately 400 million years ago (*McGeoch and Gatherer, 2005*). Homologues of pUL7 from α-, β- and γ-herpesviruses can be readily identified by their conserved amino acid sequences, despite relatively low sequence identities (HCMV and KSHV homologues share 17% and 16% identity, respectively, with HSV-1 pUL7). The predicted secondary structures of pUL7 homologues from representative α-, β- and γ-herpesviruses that infect humans are very similar to the experimentally-determined secondary structure of HSV-1 pUL7, strongly suggesting that these proteins will adopt the CUSTARD fold (*Figure 1—figure supplement 2*). Similarly, the predicted secondary structures of putative β- and γ-herpesvirus pUL51 homologues closely match the prediction for HSV-1 pUL51 (*Figure 1—figure supplement 2*) despite low sequence identity (HCMV and KSHV homologues sharing 16% and 13% identity, respectively, with HSV-1 pUL51). As the pUL7 and pUL51 homologues conserve secondary structure and, where tested, conserve function in promoting virus assembly, we sought to determine whether the formation of a pUL7: pUL51 complex is conserved across the α-, β- and γ-herpesvirus families.

GFP-tagged pUL7 homologues from human herpesviruses HSV-1, VZV, HCMV or KSHV were co-transfected with mCherry-tagged pUL51 homologues from the same virus into human embryonic kidney (HEK) 293 T cells. In all cases, pUL51-mCherry or the relevant homologue could be readily co-precipitated with the GFP-pUL7 homologue, whereas pUL51-mCherry homologues were not efficiently co-precipitated by GFP alone (*Figure 3A*). The association of pUL7 and pUL51 homologues is therefore conserved across the herpesvirus families.

Given the large evolutionary distance between α-, β- and γ-herpesvirus pUL7 and pUL51 homologues, and consequent sequence divergence, it was unclear whether the molecular details of the interaction between these proteins would be conserved. GFP-tagged pUL7 was thus co-transfected with mCherry-tagged pUL51 from HSV-1 or with mCherry-tagged homologues from VZV, HCMV or KSHV. Co-precipitation was observed for HSV-1 pUL51 and for pORF7 from VZV, an α-herpesvirus, but not for the homologues from HCMV or KSHV (*Figure 3B*). This suggested that the pUL7:pUL51 molecular interface is partially conserved within the α-herpesvirus family, but not across families. VZV pORF53 and pORF7 share 33% and 35% identity with HSV-1 pUL7 and pUL51, respectively. Mapping the conservation of α-herpesvirus pUL7 sequences onto the HSV-1 pUL7 structure reveals several regions of conservation that overlap with the binding footprint in pUL51 in the core heterodimeric complex (*Figure 3E*). However, capture of pUL51-mCherry did not result in co-precipitation of the VZV pUL7 homologue pORF53, nor did capture of GFP-pORF53 result in co-

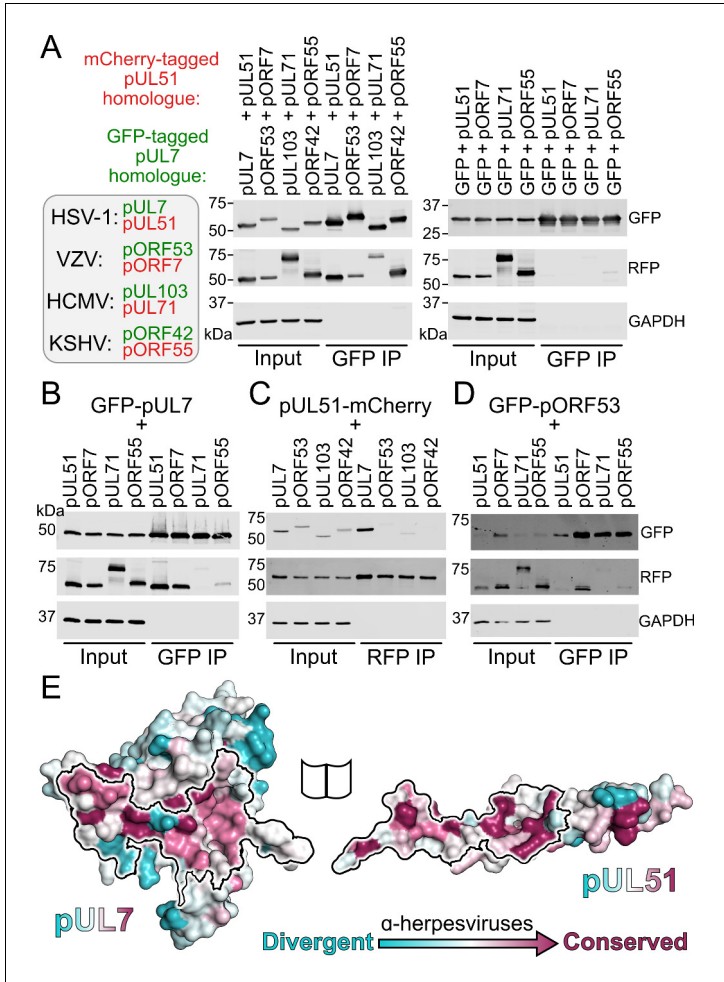

**Figure 3.** Conservation of the pUL7:pUL51 interaction across herpesviruses. (**A–D**) HEK 293 T cells were co-transfected with GFP-tagged pUL7 homologues from human herpesviruses, or with GFP alone, and with mCherry tagged pUL51 homologues. Cells were lysed 24 hr post-transfection and incubated with anti-GFP (**A, B, D**) or anti-RFP (**C**) resin to capture protein complexes before being subjected to SDS-PAGE and immunoblotting using the antibodies shown. All immunoblots are representative of at least two independent experiments performed by different scientists. (**A**) mCherry-tagged homologues of pUL51 are captured by GFP-pUL7 homologues, but not by GFP alone. (**B**) GFP-pUL7 co-precipitates with pUL51 (HSV-1) and pORF7 (VZV), but not with pUL71 (HCMV) or pORF55 (KSHV). (**C**) pUL51-mCherry co-precipitates with pUL7 but not with homologues from other herpesviruses. (**D**) The VZV pUL7 homologue pORF53 co-precipitates with VZV pORF7, but not with pUL51 homologues from other herpesviruses. (**E**) Molecular surfaces of the pUL7 and pUL51 core heterodimer, colored by residue conservation across the α-herpesviruses. Residues that mediate the pUL7:pUL51 interaction are outlined.
The online version of this article includes the following figure supplement(s) for figure 3:

**Figure supplement 1.** pUL51 does not co-precipitate pUL14 in uninfected cultured cells.

precipitation of HSV-1 pUL51 (*Figure 3C and D*). We therefore conclude that, while the pUL7:pUL51 interface is partially conserved across α-herpesviruses, there has been co-evolution of pUL7 and pUL51 homologues such that the interaction interfaces are distinct at a molecular level.

To test whether the core heterodimeric pUL7:pUL51 interaction interface is subject to co-evolutionary change, a matrix of 63 interacting pairs of residues (one from each protein) was generated by manual inspection of the binding interface. The amino acids carried at these sites across an alignment of pUL7 and pUL51 homologues from 199 strains of α-herpesvirus were tested for correlated changes. Initially, 35 of the 63 interacting-residue pairs where homology could be confidently assigned were analyzed, results being compared to a null distribution determined from $10^6$ data sets where interacting sites were paired at random. True pairings showed more correlated change than

94% of the randomized pairings and the results were little changed when different subsets of the data, including fewer strains and more interactions, were analyzed (*Supplementary file 1*–Table S4). This is suggestive evidence for co-evolution of the interaction interface across the α-herpesviruses. Similar analysis was attempted to probe for co-evolution of the core pUL7:pUL51 interaction interface across all herpesviruses, but the extensive sequence divergence confounded the confident assignment of interacting amino acid pairs (only 12 pairs could be confidently assigned) and so the subsequent analysis was underpowered.

In addition to interacting with pUL7, it has previously been shown that HSV-1 pUL51 is able to interact with HSV-1 pUL14 (*Oda et al., 2016*) and that mutation of pUL51 amino acids Ile111, Leu119 and Tyr123 to alanine disrupts this interaction. Residues Leu119 and Tyr123 are completely buried in the interface between pUL7:pUL51 in the core heterodimer structure, interacting with residues from pUL51 helix α1 and from pUL7 helices α8 and α9 (*Figure 3—figure supplement 1A*). Such burial would preclude simultaneous binding of these residues to pUL7 and pUL14. However, the second copy of pUL51 in the solution heterotrimer may be capable of binding pUL14, or pUL14 may compete with pUL7 for binding to pUL51. To test these hypotheses, pUL51-mCherry was co-transfected into mammalian cells together with GFP-pUL7 and/or myc-pUL14 and then captured using mCherry affinity resin. While GFP-pUL7 was readily co-precipitated, we could not detect co-precipitation of myc-pUL14 with pUL51-mCherry either in the presence or absence of GFP-pUL7 (*Figure 3—figure supplement 1B*). As the pUL51:pUL14 interaction was previously demonstrated using infected cells or infected-cell lysates (*Oda et al., 2016*) it seems likely that this interaction is not direct, but is instead mediated by other herpesvirus proteins and that it may require binding of pUL51 to pUL7.

## Association of pUL7:pUL51 homologues to trans-Golgi membranes is conserved but association with focal adhesions is not

In addition to the roles of the pUL7 and pUL51 in promoting virus assembly, which appear to be conserved across herpesviruses, the HSV-1 pUL7:pUL51 complex has been shown to interact with focal adhesions to stabilize the attachment of cultured cells to their substrate during infection (*Albecka et al., 2017*). To probe whether focal adhesion binding is a conserved property of pUL7:pUL51 homologues, GFP-tagged pUL7 and mCherry-tagged pUL51 (or homologous complexes) were co-transfected into HeLa cells. As previously observed, HSV-1 pUL7:pUL51 complex co-localizes with both TGN46, a *trans*-Golgi marker, and with paxillin and zyxin at the cell periphery, markers of focal adhesions (*Figure 4*; *Figure 4—figure supplement 1*; *Figure 4—figure supplement 2*). VZV pORF53:pORF7, HCMV pUL103:pUL71 and KSHV pORF42:pORF55 all co-localize with TGN46 at *trans*-Golgi membranes (*Figure 4*). However, these homologues do not co-localize with paxillin or zyxin at focal adhesions (*Figure 4—figure supplement 1*; *Figure 4—figure supplement 2*).

## pUL51 resembles cellular ESCRT-III components

While the pUL7 CUSTARD fold has not been observed previously, frustrating attempts to infer function by analogy, the helix-turn-helix fold of pUL51 residues 41–125 is a common feature of many proteins. Of the proteins identified by structure-based searches, the similarity to human CHMP4B, a component of the ESCRT-III membrane-remodeling machinery, is particularly notable given the role of pUL51 and homologues in stimulating virus wrapping (*Albecka et al., 2017*; *Jiang et al., 2017*; *Klupp et al., 2005*; *Schauflinger et al., 2011*). CHMP4B and homologues are required for the efficient fusion of membrane leaflets during vesicle budding into organelle lumens, cytokinetic abscission, nuclear envelope closure, and budding of some enveloped viruses (*McCullough et al., 2018*). Helices α1 and α2 of pUL51 superpose onto human CHMP4B (*Martinelli et al., 2012*) with 1.2 Å root-mean-squared deviation across 59 $C^{\alpha}$ atoms (*Figure 5A*). pUL51 also resembles the structures of yeast and fly CHMP4B homologues Snf7 (*Tang et al., 2015*) and Shrub (*McMillan et al., 2016*), and pUL51 can be superposed onto either structure with 1.5 Å root-mean-squared deviation across 57 $C^{\alpha}$ atoms (*Figure 5B and C*).

A conserved feature of cellular ESCRT-III components like CHMP4B is their ability to form filaments that line the neck of nascent membrane buds and act in concert with VPS4 to promote membrane scission (*McCullough et al., 2018*; *Maity et al., 2019*). Formation of such filaments is accompanied by a conformational switch from a closed, auto-inhibited form to an open,

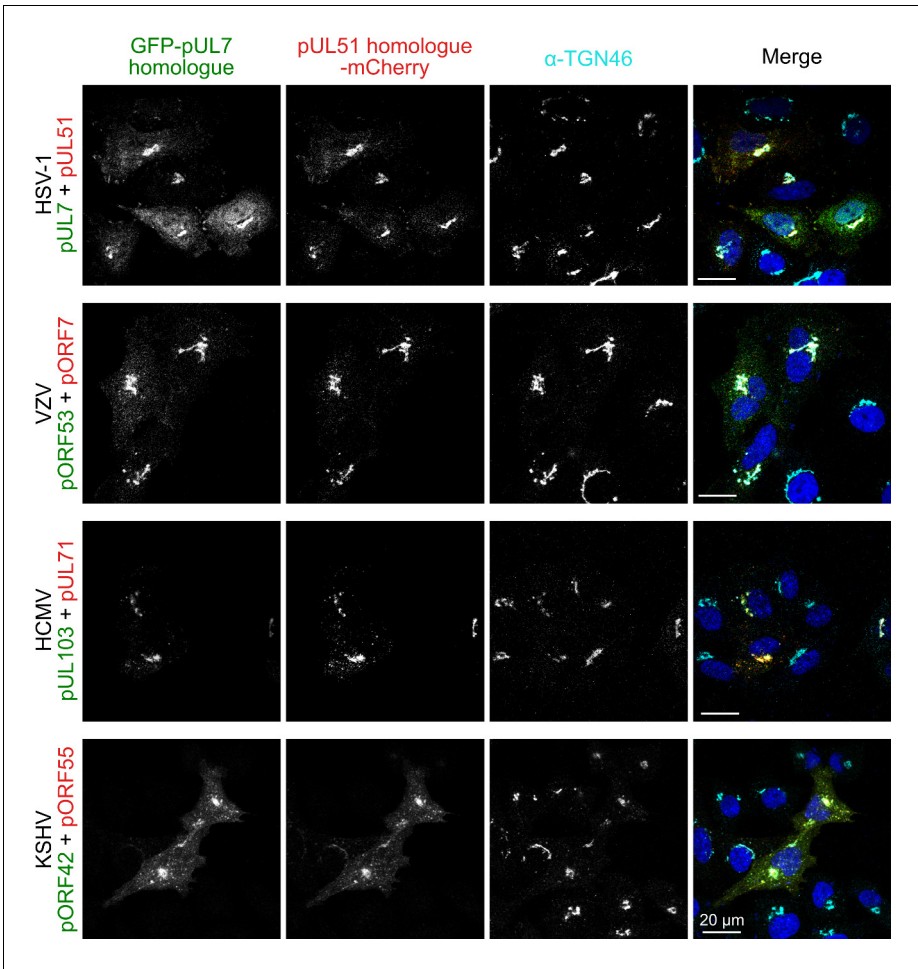

**Figure 4.** Co-localization of the pUL7:pUL51 complex with *trans*-Golgi membranes is conserved across human herpesviruses. HeLa cells were co-transfected with GFP-pUL7 and pUL51-mCherry, or with similarly-tagged homologues from VZV, HCMV and KSHV. Cells were fixed 24 hr post transfection and immunostained using the *trans*-Golgi marker protein TGN46 before imaging by confocal microscopy. Co-localization between the GFP, mCherry and far-red (TGN) fluorescence is observed in cells transfected with either HSV-1 pUL7:pUL51 or with the homologous complexes from VZV, HCMV and KSHV. HSV-1 pUL7 and pUL51 also co-localize with striated cell peripheral structures (focal adhesions, see *Figure 4—figure supplement 1* and *Figure 4—figure supplement 2*). Images are representative of experiments performed in three cell lines (TERT-immortalized human foreskin fibroblasts, U2-OS osteosarcoma cells and HeLa cells) by two independent scientists.

The online version of this article includes the following figure supplement(s) for figure 4:

**Figure supplement 1.** The pUL7:pUL51 complex from HSV-1 co-localizes with focal adhesion marker paxillin, but homologues from other human herpesviruses do not.

**Figure supplement 2.** The pUL7:pUL51 complex from HSV-1 co-localizes with focal adhesion marker zyxin but homologues from other human herpesviruses do not.

polymerization-competent form where helix α3 of the ESCRT-III core domain is continuous with helix α2 (*Tang et al., 2015*; *McMillan et al., 2016*; *McCullough et al., 2015*). The region of pUL51 immediately following helix α2 is predicted to be helical (*Figure 1—figure supplement 2*). We therefore sought to investigate whether pUL51 can form ESCRT-III–like filaments. As the C-terminal region of ESCRT-III components promote stabilization of the closed, auto-inhibited form (*Bajorek et al., 2009*; *Henne et al., 2012*), we used a truncated form of pUL51 spanning residues 1–170 that is predicted to be largely α-helical in nature (*Figure 5D*; *Figure 1—figure supplement 2*). When expressed in *E. coli* in the absence of pUL7 this protein was insoluble (*Figure 5—figure supplement 1A*). However, the protein could be readily purified from inclusion bodies and refolded in vitro by

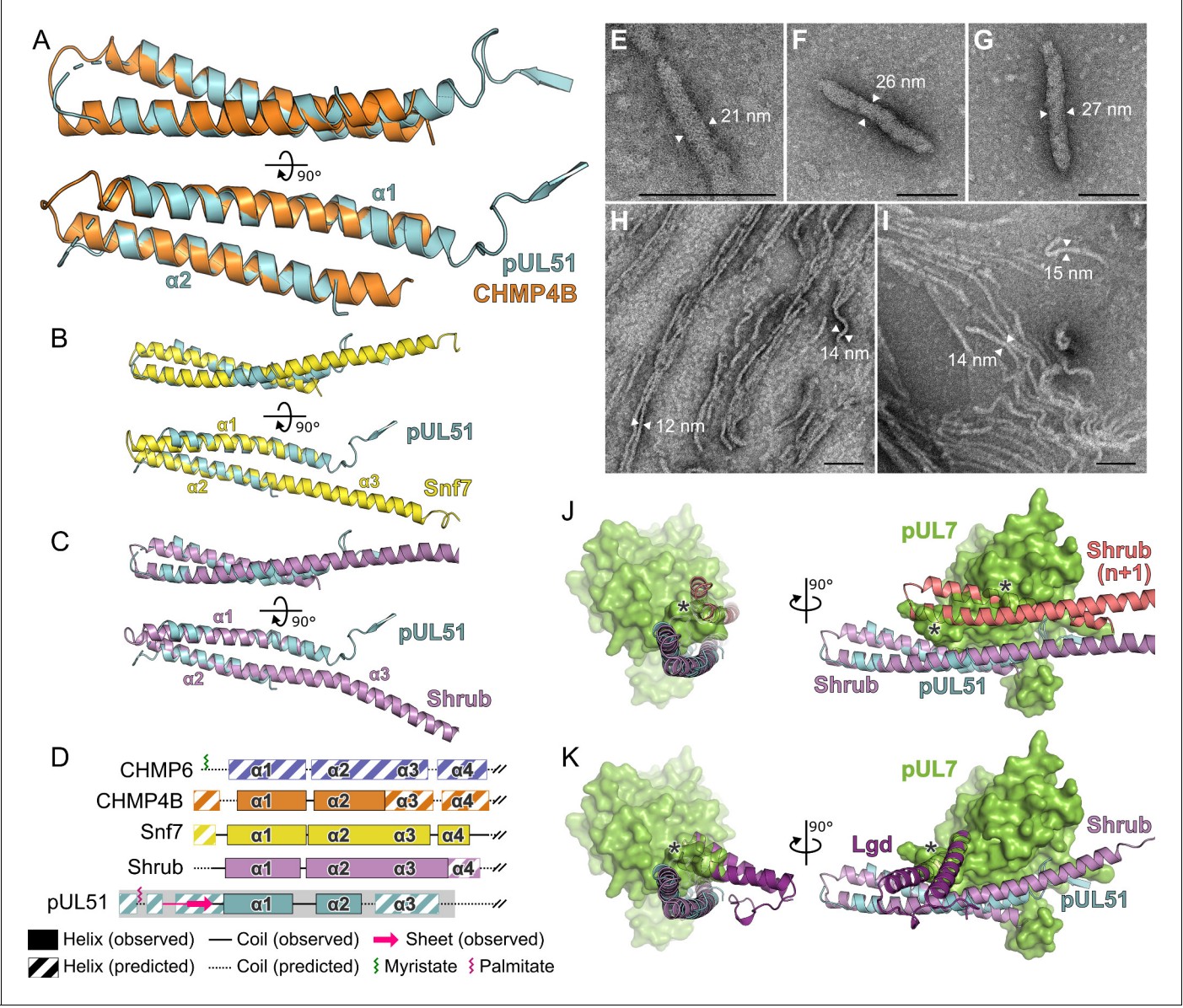

**Figure 5.** Structural similarity of HSV-1 pUL51 to cellular ESCRT-III proteins. (A) pUL51 (cyan) is shown superposed on the helical hairpin (conserved helices α1 and α2) of human CHMP4B (orange; PDB 4ABM) (*Martinelli et al., 2012*). (B and C) pUL51 (cyan) superposed on conserved helices α1 and α2 of CHMP4B homologues (B) yeast Snf7 (yellow; PDB 5FD7) (*Tang et al., 2015*) and (C) fly Shrub (violet; PDB 5J45) (*McMillan et al., 2016*). Note helices α2 and α3 of the ESCRT-III core domains of that Snf7 and Shrub are elongated and continuous in polymeric forms of these proteins (*Tang et al., 2015*; *McMillan et al., 2016*). (D) Schematic representation of selected cellular ESCRT-III protein core domains and pUL51. Residues 1–150 of cellular ESCRT-III proteins and 1–190 of pUL51 are depicted. Secondary structure of crystal structures shown in panels A–C are in solid lines (coil) and solid boxes (helices). Predicted secondary structure (*Petersen et al., 2009*) outside these regions is shown as dotted lines (coil) and striped boxes (helices). The N-terminal region of pUL51 that forms a β-sheet with the pUL7 cloning tag, presumably an artefact of crystallization, and preceding residues are shown in pink. The region of pUL51 used for electron microscopy analysis is shaded in grey. Myristoylation (CHMP6) or palmitoylation (pUL51) sites are indicated by green and purple sticks, respectively. (E–I) Negative stain transmission electron microscopy images of pUL51 filaments. Scale bars, 100 nm. (E–G) Representative images of pUL51 proto-filaments formed when 100 μM pUL51(1–170) in 20 mM tris pH 8.5 was incubated on grids for 30 s before staining. (H, I) Representative images of pUL51 filaments formed when 10 μM pUL51(1–170) in 20 mM HEPES pH 7.5 was incubated on grids for 1–2 min before staining. (J and K) The pUL7:pUL51 core heterodimer is shown superposed onto (J) two subunits of the putative Shrub homopolymer (violet and pink; PDB 5J45) (*McMillan et al., 2016*), or (K) the complex of Shrub with the regulatory DM14-3 domain of Lgd (purple; PDB 5VO5) (*McMillan et al., 2017*). pUL7 is shown as a green molecular surface. Spatial overlap between pUL7 and (J) the second subunit of Shrub, or (K) the Lgd DM14-3 domain, is denoted by asterisks.

The online version of this article includes the following figure supplement(s) for figure 5:

**Figure supplement 1.** Purification of His-tagged pUL51(1-170).

rapid dilution (*Figure 5—figure supplement 1A*). Circular dichroism spectroscopy of the refolded protein confirmed it to be largely α-helical (*Figure 5—figure supplement 1B*). While the refolded protein was soluble at low salt concentrations (≤200 mM), it rapidly aggregated to form visible precipitates at higher salt concentrations. Negative stain electron microscopy analysis of the pUL51(1–170) protein prepared in buffers lacking salt showed it to form filaments in vitro. The form of these filaments varied with pH, concentration and incubation time on the electron microscopy grid: Short proto-filaments with diameters of 20–28 nm were formed by 100 µM pUL51(1–170) at pH 8.5 in the absence of salt incubated on grids for 30 s before staining (*Figure 5E–G*), whereas longer 12–15 nm wide filaments were formed by 10 µM pUL51(1–170) in pH 7.5 HEPES incubated on grids for 1–2 min before staining (*Figure 5H and I*). These pUL51(1–170) filaments resemble the filaments observed in vitro for purified Snf7 (*Henne et al., 2012*) and CHMP4B (*Pires et al., 2009*). Therefore, in addition to sharing structural similarity to cellular ESCRT-III components, pUL51 shares the ability to form filaments in vitro.

## Discussion

We present here the structure of HSV-1 pUL7 in complex with pUL51. In solution this complex forms a 1:2 heterotrimer that is capable of forming higher-order oligomers (*Figure 1*). The C-terminal region of pUL51 is predicted to be disordered (*Figure 1—figure supplement 2*) and is extended in solution (*Figure 1C,D and L*), consistent with this region having little intrinsic structure. The crystal structure of pUL7 in complex with pUL51(8–142) shows pUL7 to comprise a single compact globular domain that adopts a previously-unobserved CUSTARD fold (*Figure 2*; *Figure 2—figure supplement 2*). A single molecule of pUL51 is bound to pUL7 in this crystal structure via an extended hydrophobic interface that is largely conserved across α-herpesviruses (*Figure 3*) and there is evidence that residues at the interface are co-evolving (*Supplementary file 1*–Table S4). Most of the pUL7-interacting residues lie within the hydrophobic loop and helix α1 of pUL51 (residues 45–88), consistent with a recent report that pUL51 residues 30–90 are sufficient for the interaction with pUL7 in transfected cells (*Feutz et al., 2019*). Recruitment of the second copy of pUL51 to the pUL7: pUL51 complex in solution requires pUL51 residues 8–40 (*Figure 1*; *Figure 2—figure supplement 1*), consistent with observations that the equivalent N-terminal region of the HCMV pUL51 homologue pUL71 is required for its self-association both in vitro and in cultured cells (*Meissner et al., 2012*), and that VZV pUL51 homologue pORF7 can also form higher-order oligomers (*Wang et al., 2017*). Furthermore, we showed that pUL51(1–170) can form long filaments that are reminiscent of those formed by cellular ESCRT-III components (*Figure 5*).

The interaction between pUL7 and pUL51 homologues is conserved across all three families of herpesvirus (*Figure 3A*), as is the association of these complexes with *trans*-Golgi compartments in cultured cells (*Figure 4*), but of the complexes tested only HSV-1 pUL7:pUL51 associates with focal adhesions in cultured cells (*Figure 4—figure supplement 1*; *Figure 4—figure supplement 2*). The conserved association of pUL7:pUL51 complexes with *trans*-Golgi membranes is consistent with a conserved role for this complex in herpesvirus assembly. Assembly of HSV-1 occurs at juxtanuclear membranes that contain cellular trans-*Golgi* and endosomal marker proteins (*Owen et al., 2015*; *Henaff et al., 2012*) and that are derived, at least in part, from recycling endosomes (*Hollinshead et al., 2012*). Similarly, HCMV assembly occurs at viral assembly compartments that contain *trans*-Golgi marker proteins (*Das et al., 2007*; *Sanchez et al., 2000*) and mutation of the pUL71 Yxxϕ motif, which mediates recycling from the plasma membrane via recognition by AP2 (*Kelly et al., 2014*), causes re-localization of pUL71 to the plasma membrane and prevents efficient HCMV assembly (*Dietz et al., 2018*). Given the conservation of the pUL7:pUL51 interaction, the conserved localization of this complex to *trans*-Golgi membranes, and the established evidence supporting roles for pUL7 or pUL51 homologues in virus assembly (*Albecka et al., 2017*; *Nozawa et al., 2005*; *Jiang et al., 2017*; *Klupp et al., 2005*; *Womack and Shenk, 2010*; *Schauflinger et al., 2011*; *Ahlqvist and Mocarski, 2011*; *Yanagi et al., 2019*; *Butnaru and Gaglia, 2019*), we propose that pUL7 and pUL51 form a complex that is conserved across herpesviruses and functions to promote virus assembly by stimulating cytoplasmic envelopment of nascent virions.

pUL51 forms large aggregates when expressed in the absence of pUL7 (*Figure 5E–G* and *Figure 1—figure supplement 1*), suggesting that the binding of pUL7 physically interferes with the ability of pUL51 to self-associate. The helix-turn-helix conformation of pUL51 resembles the cellular

ESCRT-III component CHMP4B (*Figure 5A*) and, like CHMP4B, pUL51(1–170) can form long filaments in vitro (*Figure 5H and I*). Polymerization of CHMP4B is known to be regulated by association with CC2D1A in humans (*Martinelli et al., 2012*) and in flies the protein Lgd regulates polymerization of the CHMP4B-homologue Shrub (*McMillan et al., 2017*). Superposition of the pUL7:pUL51 core heterodimer onto Shrub shows that pUL7 occupies the space that would be occupied by the adjacent Shrub molecule of a putative Shrub homopolymer (*Figure 5J*; *McMillan et al., 2016*). Similarly, the DM14-3 domain of Lgd, which is sufficient to bind Shrub in vitro and prevent Shrub polymerization (*McMillan et al., 2017*), occupies a similar space to helices α8 and α9 of pUL7 (*Figure 5K*). Taken together, these observations suggest that polymerization of pUL51 may utilize equivalent molecular surfaces as cellular CHMP4B homologues. We propose that pUL7 acts as a chaperone of pUL51, regulating its polymerization by physically inhibiting its self-association.

The N-terminal region of pUL51 is palmitoylated and this modification is required for its membrane association (*Nozawa et al., 2003*). These properties are shared by the N-terminally myristoylated cellular ESCRT-III component CHMP6 (*Yorikawa et al., 2005*). Activity of ESCRT-III components and VPS4, the AAA-ATPase that dissociates ESCRT-III and promotes bud scission (*McCullough et al., 2018*; *Maity et al., 2019*), are known to be required for efficient assembly of HSV-1 (*Crump et al., 2007*; *Pawliczek and Crump, 2009*). During cellular budding events ESCRT-III proteins are recruited to sites of membrane deformation via direct interactions with components of the ESCRT-I and ESCRT-II complexes, or with the Bro1-domain containing proteins Alix, HD-PTP or BROX (*Christ et al., 2017*). However, these proteins are not required for HSV-1 assembly (*Pawliczek and Crump, 2009*; *Barnes and Wilson, 2020*). The lack of requirement for cellular initiators of ESCRT-III polymerization, combined with the ability of pUL51 to bind directly to membranes and to form filaments, suggests that pUL51 may directly promote membrane deformation and virus budding – effectively performing the roles of multiple cellular ESCRT-III components. This proposal is consistent with observations made in HCMV, where mutation of the pUL51 homologue pUL71 results in the accumulation of HCMV particles in membrane buds with narrow necks (*Read et al., 2019*) that are reminiscent of the stalled budding profiles observed for HIV-1 when ESCRT-III activity is perturbed (*von Schwedler et al., 2003*) or HSV-1 in cells expressing a dominant negative form of VPS4 (*Crump et al., 2007*).

The mechanism by which herpesviruses recruit ESCRT-III to tegument-wrapped capsids in order to catalyze cytoplasmic envelopment remains poorly characterized (*Barnes and Wilson, 2019*). Based on the structural and functional homology between pUL51 and CHMP4B/CHMP6 we propose that pUL51 and homologues act as viral ESCRT-III components. The interaction between pUL7 and pUL51 homologues is conserved across herpesviruses, and we propose that this interaction regulates polymerization of pUL51 homologues in infected cells. It remains unclear whether there exists a trigger that would promote pUL7 dissociation from pUL51, or whether high local concentrations of pUL51 at sites of virus assembly would be sufficient to stimulate pUL51 polymerization. Furthermore, as deletion of pUL51 or its homologues does not completely abolish virus replication (*Albecka et al., 2017*; *Nozawa et al., 2005*; *Roller et al., 2014*; *Jiang et al., 2017*; *Klupp et al., 2005*; *Schauflinger et al., 2011*; *Yanagi et al., 2019*) it is likely that herpesviruses use multiple, redundant mechanisms to ensure efficient wrapping of nascent virions.

## Materials and methods

**Key resources table**

| Reagent type (species) or resource | Designation | Source or reference | Identifiers | Additional information |
|---|---|---|---|---|
| Gene (herpes simplex virus 1) | UL7 | Human herpesvirus 1 strain KOS, complete genome | GenBank: JQ673480.1; UniProt: A0A110B4Q7 | |
| Gene (herpes simplex virus 1) | UL51 | Human herpesvirus 1 strain KOS, complete genome | GenBank: JQ673480.1; UniProt: D3YPL0 | |

*Continued on next page*

Continued

| Reagent type (species) or resource | Designation | Source or reference | Identifiers | Additional information |
|---|---|---|---|---|
| Gene (varicella zoster virus) | ORF53 | Human herpesvirus 3 (HHV-3), complete genome, isolate HJ0 | GenBank: AJ871403.1; Uniprot: P09301 | |
| Gene (varicella zoster virus) | ORF7 | Human herpesvirus 3 (HHV-3), complete genome, isolate HJ0 | GenBank: AJ871403.1; Uniprot: P09271 | |
| Gene (human cytomegalovirus) | UL103 | Human herpesvirus 5 strain Toledo, complete genome | GenBank: GU937742.2; Uniprot: D3YS25 | |
| Gene (human cytomegalovirus) | UL71 | Human herpesvirus 5 strain Toledo, complete genome | GenBank: GU937742.2; Uniprot: D3YRZ9 | |
| Gene (Kaposi's sarcoma-associated herpesvirus) | ORF42 | Human herpesvirus 8 strain JSC-1 clone BAC16, complete genome | GenBank: GQ994935.1; Uniprot: F5HAI6 | |
| Gene (Kaposi's sarcoma-associated herpesvirus) | ORF55 | Human herpesvirus 8 strain JSC-1 clone BAC16, complete genome | GenBank: GQ994935.1; Uniprot: F5H9W9 | |
| Strain, strain background (Escherichia coli) | T7 express lysY/I^q | New England BioLabs | Cat#: C2566H | |
| Cell line (Homo sapiens) | HEK 293 T | ATCC | Cat#: CRL-3216; RRID:CVCL_0063 | |
| Cell line (Homo sapiens) | HeLa | ATCC | Cat#: CCL-2; RRID:CVCL_0030 | |
| Antibody | Anti-GFP (rabbit polyclonal) | Merck | Cat#: G1544; RRID:AB_439690 | (1:5000) |
| Antibody | Anti-RFP (rat monoclonal) | Chromotek | Cat#: 5F8; RRID:AB_2336064 | (1:1000) |
| Antibody | Anti-GAPDH (mouse monoclonal) | ThermoFisher | Cat#: AM4300; RRID:AB_2536381 | (1:200,000) |
| Antibody | IRDye 680T conjugated goat anti-rat (polyclonal) | LI-COR | Cat#: 926–68029; RRID:AB_10715073 | (1:10,000) |
| Antibody | IRDye 680T conjugated donkey anti-rabbit (polyclonal) | LI-COR | Cat#: 926–68023; RRID:AB_10706167 | (1:10,000) |
| Antibody | IRDye 680T conjugated goat anti-mouse (polyclonal) | LI-COR | Cat#: 926–68020; RRID:AB_10706161 | (1:10,000) |
| Antibody | IRDye 800CW conjugated donkey anti-rabbit (polyclonal) | LI-COR | Cat#: 926–32213; RRID:AB_621848 | (1:10,000) |
| Antibody | IRDye 800CW conjugated goat anti-mouse (polyclonal) | LI-COR | Cat#: 926–32210; RRID:AB_621842 | (1:10,000) |
| Antibody | Anti-TGN46 (sheep polyclonal) | Bio-Rad | Cat#: AHP500G; RRID:AB_323104 | (1:200) |

*Continued*

| Reagent type (species) or resource | Designation | Source or reference | Identifiers | Additional information |
|---|---|---|---|---|
| Antibody | Anti-paxillin (mouse monoclonal) | BD Biosciences | Cat#: 610051; RRID:AB_397463 | (1:300) |
| Antibody | Anti-zyxin (rabbit polyclonal) | Abcam | Cat#: ab71842; RRID:AB_2221280 | (1:100) |
| Antibody | Alexa Fluor 647 conjugated anti-sheep (donkey polyclonal) | ThermoFisher | Cat#: A-21448; RRID:AB_2535865 | (1:1000) |
| Antibody | Alexa Fluor 647 conjugated anti-mouse (goat polyclonal) | ThermoFisher | Cat#: A-21236; RRID:AB_2535805 | (1:1000) |
| Antibody | Alexa Fluor 647 conjugated anti-rabbit (goat polyclonal) | ThermoFisher | Cat#: A-21245; RRID:AB_2535813 | (1:200) |
| Recombinant DNA reagent | His-pUL51 | (*Albecka et al., 2017*) | | |
| Recombinant DNA reagent | His-pUL51 C9S | This paper | | Generated by site-directed mutagenesis of His-UL51(FL) to substitute Cys9 with serine |
| Recombinant DNA reagent | His-pUL51(1-170) | This paper | | Residue Cys9 was substituted with serine |
| Recombinant DNA reagent | UL7-GST:pUL51 | This paper | | pUL51 residue Cys9 was substituted with serine; Codon optimised pUL7 (GeneArt) |
| Recombinant DNA reagent | UL7-GST: UL51(8-142) | This paper | | pUL51 residue Cys9 was substituted with serine; Codon optimised pUL7 (GeneArt) |
| Recombinant DNA reagent | GST-UL7: UL51(8-142) | This paper | | pUL51 residue Cys9 was substituted with serine; Codon optimised pUL7 (GeneArt) |
| Recombinant DNA reagent | UL7-GST: UL51(41-142) | This paper | | Codon optimised pUL7 (GeneArt) |
| Recombinant DNA reagent | GFP-pUL7 | (*Albecka et al., 2017*) | | |
| Recombinant DNA reagent | pUL51-mCherry | (*Albecka et al., 2017*) | | |
| Recombinant DNA reagent | GFP-pORF53 | This paper | | Codon optimized (GeneArt) |
| Recombinant DNA reagent | pORF7-mCherry | This paper | | Codon optimized (GeneArt) |
| Recombinant DNA reagent | GFP-pUL103 | This paper | | |
| Recombinant DNA reagent | pUL71-mCherry | This paper | | |

*Continued on next page*

*Continued*

| Reagent type (species) or resource | Designation | Source or reference | Identifiers | Additional information |
|---|---|---|---|---|
| Recombinant DNA reagent | GFP-pORF42 | This paper | | |
| Recombinant DNA reagent | pORF55-mCherry | This paper | | |
| Sequence-based reagent | UL51_C9S_F | This paper | Site-directed mutagenesis primer | CTCGGGGCTATAAGTGGCTGGGGAG |
| Sequence-based reagent | UL51_C9S_R | This paper | Site-directed mutagenesis primer | CTCCCCAGCCACTTATAGCCCCGAG |
| Software, algorithm | NetSurfP | Technical University of Denmark | http://www.cbs.dtu.dk/services/NetSurfP/ | Version 1.1 |
| Software, algorithm | MoreRONN | Dr Varun Ramraj and Dr Robert Esnouf, University of Oxford | | Version 4.6 |
| Software, algorithm | Astra | Wyatt Technology | RRID:SCR_016255 | Version 6 |
| Software, algorithm | CSS-Palm | The Cuckoo Workgroup | http://csspalm.biocuckoo.org/online.php | Version 4.0 |
| Software, algorithm | I-TASSER | Zhang lab | RRID:SCR_014627 | |
| Software, algorithm | PDBeFOLD | EBI | https://www.ebi.ac.uk/msd-srv/ssm/ | |
| Software, algorithm | DALI | Holm group, University of Helsinki | RRID:SCR_013433 | |
| Software, algorithm | CATHEDRAL | CATH | http://www.cathdb.info/search/by_structure | |
| Software, algorithm | Clustal Omega | EBI | RRID:SCR_001591 | |
| Software, algorithm | HMMER | HMMER | RRID:SCR_005305 | |
| Software, algorithm | ConSurf | ConSurf | RRID:SCR_002320 | |
| Software, algorithm | CCP4i2 package | CCP4i2 | RRID:SCR_007255 | Version 7.0 |
| Software, algorithm | BUSTER | BUSTER | RRID:SCR_015653 | Version 2.10.3 |
| Software, algorithm | PyMOL | PyMOL | RRID:SCR_000305 | Open source version |
| Software, algorithm | GraphPad Prism | GraphPad | RRID:SCR_002798 | Version 7 |
| Software, algorithm | Inkscape | Inkscape | RRID:SCR_014479 | Version 0.92.3 |
| Software, algorithm | ATSAS package | EMBL Hamburg | RRID:SCR_015648 | Version 2.8.4 |
| Software, algorithm | Proteome Discoverer | ThermoFisher | RRID:SCR_014477 | Version 2.2.0.388 |
| Software, algorithm | CDSSTR | CDSSTR | http://dichroweb.cryst.bbk.ac.uk/ | As implemented by DichroWeb |

## Protein production

Full-length herpes simplex virus (HSV)−1 strain KOS protein pUL51 (UniProt ID D3YPL0), either with the wild-type sequence or where residue Cys9 (the palmitoyl group acceptor) had been substituted with serine, was expressed with an N-terminal MetAlaHis$_6$ tag and purified by Ni$^{2+}$ affinity capture and size-exclusion chromatography as described in *Albecka et al., 2017*. pUL51(1-170) was expressed with an N-terminal MetAlaHis$_6$ tag and residue Cys9 substituted to serine in the *Escherichia coli* strain T7 express *lysY/I$^q$* (New England BioLabs). Bacteria were cultured in 2 × TY medium, recombinant proteins being expressed overnight at 25°C following addition of 0.4 mM isopropyl β-D-1-thiogalactopyranoside. The complex of HSV-1 strain KOS proteins pUL7 (UniProt ID A0A110B4Q7) and pUL51, or truncations thereof, were co-expressed in the *E. coli* strain T7 express *lysY/I$^q$* (New England BioLabs) using the polycistronic vector pOPC (*Tan, 2001*). The nucleotide sequence of pUL7 had been optimized to enhance recombinant expression (GeneArt) and, where present, residue Cys9 of pUL51 had been substituted with serine. For all experiments except *Figure 2—figure supplement 1*, pUL7 was fused to a C-terminal human rhinovirus 3C protease recognition sequence and GST purification tag. For *Figure 2—figure supplement 1A*, the GST and 3C recognition sequence were fused to the N terminus of pUL7. Bacteria were cultured in 2 × TY medium, recombinant proteins being expressed overnight at 22°C following addition of 0.4 mM isopropyl β-D-1-thiogalactopyranoside.

Bacterial cell pellets were resuspended in lysis buffer (50 mM sodium phosphate pH 7.5, 500 mM NaCl, 0.5 mM MgCl$_2$, 1.4 mM β-mercaptoethanol, 0.05% Tween-20) supplemented with 400 U bovine pancreas DNase I (Merck) and 200 µL EDTA-free protease inhibitor cocktail (Merck) at 4°C. Cells were lysed using a TS series cell disruptor (Constant Systems) at 24 kPSI and the lysate was cleared by centrifugation at 40,000 × g for 30 min at 4°C. For soluble proteins, cleared lysate was incubated with glutathione sepharose 4B resin (GE Healthcare) equilibrated in GST wash buffer (50 mM sodium phosphate pH 7.5, 300 mM NaCl, 1 mM dithiothreitol (DTT)) for 1 hr at 4°C before being applied to a column and washed with >10 column volumes (c.v.) of GST wash buffer. To remove contaminating nucleic acids, pUL7:pUL51 complexes were resuspended in 25 mM sodium phosphate pH 7.5, 150 mM NaCl, 0.5 mM DTT, 1 mM MgCl$_2$ and incubated with 2000 U of benzonase (Merck) for 1 hr at room temperature before being applied to a column, washed with 8 c.v. of 50 mM sodium phosphate pH 7.5, 1M NaCl, and then washed with 4 c.v. of GST wash buffer. Bound protein was eluted using GST wash buffer supplemented with 25 mM reduced L-glutathione, concentrated, and further purified by size-exclusion chromatography (SEC) using an S200 16/600 column (GE Healthcare) equilibrated in 20 mM tris pH 7.5, 200 mM NaCl, 1 mM DTT. The GST tag was removed by supplementing the pooled SEC fractions containing pUL7:pUL51 complex with 0.5 mM EDTA and then incubating with 40 µg of GST-tagged human rhinovirus 3C protease. Free GST and uncleaved GST-tagged pUL7 were captured using glutathione sepharose resin and the cleaved complex was again subjected to SEC using S200 16/600 or 10/300 columns (GE Healthcare) equilibrated in 20 mM tris pH 7.5, 200 mM NaCl, 1 mM DTT, 3% (v/v) glycerol. Purified pUL7:pUL51 was concentrated, snap-frozen in liquid nitrogen as small aliquots, and stored at −80°C. Protein concentrations were estimated from absorbance at 280 nm using calculated extinction coefficients (*Wilkins et al., 1999*) where pUL7 and pUL51 were assumed to be present in 1:2 molar ratios for all complexes except for pUL7:pUL51(41-142), where an equimolar ratio was assumed.

pUL51(1-170) was purified from inclusion bodies and refolded by rapid dilution as described previously for vaccinia virus CrmE (*Graham et al., 2007*). Briefly, cells were lysed and the lysates clarified as above. Insoluble pellets were then washed four times by resuspension in inclusion body wash buffer (50 mM tris pH 7.5, 100 mM NaCl, 0.5% Triton X-100) using a Dounce homogenizer, followed by centrifugation at 25,000 × g for 10 min at 4°C. Pellets were washed once with inclusion body wash buffer without Triton X-100, then resuspended in solubilization buffer (50 mM tris pH 7.5, 100 mM NaCl, 6 M guanidine hydrochloride, 10 mM EDTA, 10 mM DTT) for 3 hr at 4°C. Protein concentration was estimated from absorbance at 280 nm using a calculated extinction coefficient (*Wilkins et al., 1999*) and the unfolded protein was stored at −20°C. To refold, 20 mg aliquots of pUL51(1-170) were thawed and supplemented with 10 mM DTT, then subjected to a rapid 1:100 (v/v) dilution into refold buffer (200 mM tris pH 7.5, 10 mM EDTA, 1 M L-arginine, 1% (v/v) EDTA-free protease inhibitor cocktail (Merck)) that was briskly stirred for 2 hr at 4°C. Refolded pUL51(1-170) was buffer-exchanged into 20 mM phosphate buffer pH 7.5 or 20 mM HEPES pH 7.5 using a

Sephadex PD-10 gravity column (GE Healthcare) or into 20 mM tris pH 8.5 by exhaustive dialysis overnight at 4°C.

## Multi-angle light scattering

Multi-angle light scattering (MALS) experiments were performed immediately following SEC (SEC-MALS) by inline measurement of static light scattering (DAWN 8+; Wyatt Technology), differential refractive index (Optilab T-rEX; Wyatt Technology), and UV absorbance (1260 UV; Agilent Technologies). Samples (100 µL) were injected onto an S200 Increase 10/300 column (GE Healthcare) equilibrated in in 20 mM tris pH 7.5, 200 mM NaCl, 3% (v/v) glycerol, 0.25 mM tris(2-carboxyethyl) phosphine (TCEP) at 0.4 mL/min. Molecular masses were calculated using ASTRA 6 (Wyatt Technology) and figures were prepared using Prism 7 (GraphPad).

## Small-angle X-ray scattering and ab initio modelling

Continuous flow small-angle X-ray scattering (SAXS) experiments were performed immediately following SEC with in-line MALS and dynamic light scattering (SEC-SAXS-MALS-DLS), at EMBL-P12 bio-SAXS beam line (PETRAIII, DESY, Hamburg) (*Blanchet et al., 2015*; *Graewert et al., 2015*). Scattering data (I(s) versus s, where s = 4πsinθ/λ nm$^{-1}$, 2θ is the scattering angle, and λ is the X-ray wavelength, 0.124 nm) were recorded using a Pilatus 6M detector (Dectris) with 1 s sample exposure times for a total of 3600 data frames spanning the entire course of the SEC separation. 90 µL of purified pUL7:pUL51 (8 mg/mL) or pUL7:pUL51(8–142) (4.5 mg/mL) was injected at 0.5 mL/min onto an S200 Increase 10/300 column (GE Healthcare) equilibrated in 20 mM HEPES pH 7.5, 200 mM NaCl, 3% (v/v) glycerol, 1 mM DTT (pUL7:pUL51) or 20 mM tris pH 7.5, 200 mM NaCl, 3% (v/v) glycerol, 0.25 mM TCEP (pUL7:pUL51(8–142)). Data presented in *Figure 1* are representative of three replicate SEC-SAXS experiments. SAXS data for the pUL7:pUL51 complex, which eluted as two peaks, were recorded from macromolecule-containing and -free fractions as follows: heterohexamers (frames 1236–1283 s), heterotrimers (frames 1382–1416 s) and solvent blank (spanning pre- and post-sample elution frames). For the pUL7:pUL51(8–142) complex, the SEC-SAXS experiment was performed three times as follows: heterotrimers (frames 1779–1886 s, 1785–1876 s and 1781–1880 s for the three experiments, respectively) and solvent blanks (spanning pre- and post-sample elution frames). Primary data reduction was performed using CHROMIXS (*Panjkovich and Svergun, 2018*) and 2D-to-1D radial averaging was performed using the SASFLOW pipeline (*Franke et al., 2012*). Buffer frames were tested for statistical equivalence using all pairwise comparison CorMap p values set at a significance threshold (α) of 0.01 (*Franke et al., 2015*) before being averaged to generate a final buffer scattering profile and subtracted from the relevant macromolecule elution peaks. Subtracted data blocks producing a consistent $R_g$ through the elution profile (as evaluated using the Guinier approximation) (*Guinier, 1939*) were scaled and checked for similarity using CorMap before being averaged to produce the final reduced 1D scattering profiles. For the pUL7:pUL51(8–142) construct, the averaged scattering profiles obtained for the three repeated SEC-SAXS measurements underwent additional scaling and final combined averaging. Primary data processing, including all CorMap calculations, was performed in PrimusQT of the ATSAS package (*Petoukhov et al., 2012*). Molecular weight estimates were calculated using the datporod (Porod volume) (*Petoukhov et al., 2012*), datmow (*Fischer et al., 2010*), datvc (*Rambo and Tainer, 2013*) and Bayesian consensus modules (*Hajizadeh et al., 2018*) of the ATSAS package. Indirect inverse Fourier transform of the SAXS data and the corresponding probable real space-scattering pair distance distributions (p(r) versus r profile) were calculated using GNOM (*Svergun, 1992*), from which the $R_g$ and $D_{max}$ were determined. In addition, the a priori assessment of the non-uniqueness of scattering data was performed using AMBIMETER (*Petoukhov and Svergun, 2015*). SAXS data collection and analysis parameters are summarized in *Supplementary file 1*–Table S1. *Ab initio* modeling was performed using GASBOR (*Svergun et al., 2001*) and DAMMIN (*Svergun, 1999*). For pUL7:pUL51, reciprocal space intensity fitting accounting for oligomeric equilibrium with P2 symmetry imposed (GASBORMX) was used to simultaneously fit the 1:2 (heterotrimer) and 2:4 (heterohexamer) pUL7:pUL51 SAXS profiles. The two SEC-elution peaks contained heterohexamer:heterotrimer volume fractions of 1.0:0.0 and 0.2:0.8, respectively, as determined by GASBORMX. Because SAXS data can be ambiguous with respect to shape restoration, DAMMIN and GASBOR were run 20 times and the consistency of the individual models was evaluated using the normalized spatial discrepancy (NSD) metric (*Volkov and*

*Svergun, 2003*). Dummy-atom models were clustered using DAMCLUST (*Volkov and Svergun, 2003*), averaged using DAMCLUST (pUL7:pUL51) or DAMAVER (pUL7:pUL51(8–142)), and refined using DAMMIN. For the pUL7:pUL51 heterohexamer three clusters were identified, which visually corresponded to parallel or anti-parallel dimers of heterotrimers, whereas for the pUL7:pUL51(8–142) heterotrimer all models formed a single cluster. The refined dummy-atom models that best fit the SAXS profile (lowest $\chi^2$) are shown in *Figure 1*.

## Cross-linking and mass spectrometry

Purified pUL7:pUL51(8–142) at 1 mg/mL (16.4 µM) in HEPES SAXS buffer was incubated with 20- to 100-fold molar excess of disuccinimidyl sulfoxide (DSSO; ThermoFisher) or disuccinimidyl dibutyric urea (DSBU; ThermoFisher) dissolved in DMSO, or with DMSO carrier alone (the final DMSO concentration remaining below 2% (v/v) in all cases). Reactions were incubated at room temperature for 30 min before quenching by addition of 1 M tris pH 7.5 to a final tris concentration of 20 mM. Samples were separated by SDS-PAGE using a 4–12% Bolt Bis-Tris gel (ThermoFisher) in MOPS running buffer and stained with InstantBlue Coomassie Protein Stain (Expedeon) according to the manufacturers' instructions. Cross-linked samples corresponding to pUL7:pUL51(8–142) heterodimers (1:1) or heterotrimers (1:2) were excised, reduced, alkylated and digested in-gel using trypsin. The resulting peptides were analyzed using an Orbitrap Fusion Lumos coupled to an Ultimate 3000 RSLC nano UHPLC equipped with a 100 µm ID × 2 cm Acclaim PepMap Precolumn and a 75 µm ID × 50 cm, 2 µm particle Acclaim PepMap RSLC analytical column (ThermoFisher Scientific). Loading solvent was 0.1% formic acid (FA) with analytical solvents A: 0.1% FA and B: 80% (v/v) acetonitrile (MeCN) + 0.1% FA. Samples were loaded at 5 µL/min, loading solvent for 5 min before beginning the analytical gradient. The analytical gradient was 3% to 40% B over 42 min, rising to 95% B by 45 min, followed by a 4 min wash at 95% B, and finally equilibration at 3% solvent B for 10 min. Columns were held at 40°C. Data were acquired in a DDA fashion with MS3 triggered by a targeted mass difference. MS1 was acquired from 375 to 1500 Th at 60,000 resolution, $4 \times 10^5$ AGC target and 50 ms maximum injection time. MS2 used quadrupole isolation at an isolation width of m/z 1.6 and CID fragmentation (25% NCE). Fragment ions were scanned in the Orbitrap with $5 \times 10^4$ AGC target and 100 ms maximum injection time. MS3 was triggered by a targeted mass difference of 25.979 for DSBU and 31.9721 for DSSO with HCD fragmentation (30% NCE) and fragment ions scanned in the ion trap with an AGC target of $2.0 \times 10^4$.

Raw files were process using XLinkX 2.2 in Proteome Discoverer 2.2.0.388 (ThermoFisher). MS2 or MS3 spectra were selected based on the identification of either DSSO (K +158.004 Da) or DSBU (K +196.085 Da) and then processed in two workflows in parallel with the following parameters. Workflow 1: XlinkX Search against a database containing an HSV-1 proteome (downloaded 04.04.2016), *E. coli* proteome (downloaded 06.09.2019 with OPGE removed) and 246 common contaminants; full trypsin digestion; carbamidomethyl static modification of cysteines; oxidation variable modification of methionines; 1% FDR using XlinkX validator Percolator. Workflow 2: spectra filtered for either MS2 or MS3 scans with each set searched separately using Mascot against a database containing an *E. coli* proteome (downloaded 06.09.2019 with OPGE removed) with 246 common contaminants, and HSV-1 proteome (downloaded 04.04.2016); PSM validator Max. Delta Cn = 0.05. Statistical validation of identified cross-link peptides from both workflows was carried out by a joint consensus workflow.

## Pseudo-atomic modelling of pUL7:pUL51(8–142) heterotrimer SAXS profile

Pseudo-atomic modelling of the pUL7:pUL51(8–142) heterotrimer was performed using CORAL (*Petoukhov et al., 2012*). The core heterodimer structure, comprising pUL7 and pUL51(41–125), was fixed in this model and a second copy of pUL51(41–125) was free to move. To include a priori information about predicted secondary structure (*Figure 1—figure supplement 2*), the pUL51(8–142) sequence was modelled by I-TASSER (*Yang et al., 2015*) using pUL51 residues 41–125 from core heterodimer structure as a template. Secondary structural (helical) elements from the I-TASSER model were included for regions of pUL51 that were disordered in the crystal structure (residues 8–23 and 126–142) or involved in the artefactual interaction with the pUL7 purification tag (residues 24–40). DSSO and DSBU cross-links were used to generate maximal inter-residue distance restraints

of 26.1 and 28.3 Å, respectively (*Merkley et al., 2014*). Cross-links between residues of pUL7 and pUL51 that are not feasible based on the core heterodimer structure were assumed to be between pUL7 and the additional copy of pUL51. Cross-links that could not be assigned unambiguously (e.g. cross-links between pUL51 residues that could be either inter- or intra-molecular) were permuted and all possible restraint geometries were tested by modelling against the pUL7:pUL51(8–142) SAXS profile (s < 3.2 nm$^{-1}$). The final distribution of target function (F) values was clearly bimodal: models from the cluster with higher F values were unable to simultaneously satisfy the provided cross-link restraints and the SAXS data, and were thus discarded. Remaining models were assessed for fit to the SAXS profile ($\chi^2$) using CRYSOL.

## X-ray crystallography

pUL7:pUL51(8–142) was crystallized in sitting or hanging drops by mixing 1 µL of 5.3 mg/mL protein with 0.5 µL of 0.5 M benzamidine hydrochloride and 1 µL of reservoir solution containing 0.15 M sodium citrate pH 5.5, 12% (v/v) 2-methyl-2,4-pentanediol, 0.1 M NaCl and equilibrating against 200 µL reservoirs at 16°C for at least one week. Crystals of pUL7:pUL51(8–142) were cryoprotected by brief immersion in reservoir solution supplemented with 20% (v/v) glycerol before flash cryo-cooling by plunging into liquid nitrogen. For multiple-wavelength anomalous dispersion (MAD) phasing experiments, 1 µL of 1 mM mercury(II) acetate in reservoir solution was added to the mother liquor and incubated at 16°C for 4 hr before cryoprotection and cryo-cooling as described above. Diffraction data were recorded at 100 K on a Pilatus3 6M detector (Dectris) at Diamond Light Source beamline I03. Images were processed using DIALS (*Winter et al., 2018*), either using the DUI graphical interface (*Fuentes-Montero et al., 2016*) for the native dataset or the xia2 automated processing pipeline (*Winter, 2010*) for the mercury derivative datasets. Scaling and merging was performed using AIMLESS (*Evans and Murshudov, 2013*) and data collection statistics are shown in *Supplementary file 1*–Table S2.

Four-wavelength anomalous dispersion analysis of the mercury derivative (space group *P* 4 2$_1$ 2) was performed using CRANK2 (*Skubák and Pannu, 2013*), followed by iterative density modification and automated model building using parrot (*Cowtan, 2010*) and buccaneer (*Cowtan, 2012*; *Cowtan, 2006*), part of the CCP4 program suite (*Winn et al., 2011*). An initial model comprising a single pUL7:pUL51(8–142) core heterodimer was used as a molecular replacement model to solve the structure of the native complex (space group *P* 2$_1$) using MolRep (*Vagin and Teplyakov, 2010*), identifying four core heterodimers in the asymmetric unit with *pseudo* four-fold non-crystallographic symmetry. Density modification and automated model building were performed using parrot and buccaneer, respectively, followed by cycles of iterative manual rebuilding in COOT (*Emsley et al., 2010*) and TLS plus positional refinement using Refmac5 (*Murshudov et al., 1997*) with local NCS restraints. The building was assisted by the use of real-time molecular dynamics-assisted model building and map fitting with the program ISOLDE (*Croll, 2018*). Final cycles of refinement following manual rebuilding were performed using autoBUSTER (*Bricogne et al., 2017*) with local NCS restraints and TLS groups that were identified with the assistance of the TLSMD server (*Painter and Merritt, 2006*). The quality of the model was monitored throughout the refinement process using MolProbity (*Chen et al., 2010*) and the validation tools in COOT. Molecular graphics were produced using PyMOL (*Schrodinger LLC, 2015*). Conservation of pUL7 and pUL51 residues across the α-herpesviruses was mapped onto the structure using the CONSURF server (*Ashkenazy et al., 2016*) and the sequence alignment used for co-evolutionary analysis (*Data set 2*, below).

## Circular dichroism spectropolarimetery

Circular dichroism spectra were recorded on a Jasco J-810 spectropolarimeter at 20°C using 1 mg/mL pUL51(1–170) in 20 mM phosphate buffer, pH 7.5. A total of 20 spectra were recorded per sample at 50 nm/min with 1 nm bandwidth between 260–190 nm. Spectra were converted to mean residue ellipticity, averaged, and smoothed (Savitzky and Golay method, second order smoothing, 5 nm sliding window) using Prism 7 (GraphPad). Spectra were decomposed using CDSSTR (*Sreerama and Woody, 2000*) as implemented by DichroWeb (*Whitmore and Wallace, 2008*) using a 1 nm wavelength step and reference set 7.

## Negative stain transmission electron microscopy

Copper grids (300 mesh) coated with formvar and continuous carbon (EM Systems Support) were glow discharged in air for 20 s. Three microlitres of 10–100 µM pUL51(1-170) in 20 mM HEPES pH 7.5 or 20 mM tris pH 8.5 was applied to the grid and allowed to adsorb (30 s to 2 min) before wicking away excess solvent with filter paper (Whatman). Grids were sequentially applied to two 30 µL drops of 2% (w/v) uranyl acetate for approximately 3 s and then 30 s, respectively. Excess stain was wicked away using filter paper (Whatman) and grids were allowed to air dry. Images were obtained using a Tecnai Spirit transmission electron microscope (FEI) operating at 120 kV, equipped with an Ultrascan 1000 CCD camera (Gatan). Images were acquired at 30,000–120,000 × magnification with −1 µm defocus and a total electron dose of 20–40 e⁻/A² across 1 s exposures.

## Bioinformatics and evolutionary analysis

Protein sequences of pUL7 and pUL51 homologues from representative α-, β- and γ-herpesviruses that infect humans were as follows (Uniprot ID): HSV-1 pUL7 (A0A110B4Q7) and pUL51 (D3YPL0), VZV pORF53 (P09301) and pORF7 (P09271), HCMV pUL103 (D3YS25) and pUL71 (D3YRZ9), human herpesvirus 7 (HHV-7) U75 (P52458) and U44 (P52474), KSHV pORF42 (F5HAI6) and pORF55 (F5H9W9), Epstein-Barr virus (EBV) BBRF2 (P29882) and BSRF1 (P0CK49). Secondary structure prediction was performed using the NetSurfP-1.1 server (*Petersen et al., 2009*), disorder prediction was performed using moreRONN version 4.6 (*Ramraj, 2014*) and palmitoylation sites were predicted using CSS-Palm 4.0 (*Ren et al., 2008*) using the confidence threshold 'High'. Structure-based database searches for proteins with similar folds to pUL7 or pUL51 were performed using PDBeFOLD (*Krissinel and Henrick, 2004*), DALI (*Holm and Laakso, 2016*) and CATHEDRAL (*Redfern et al., 2007*).

Clustal Omega (*Sievers and Higgins, 2018*) was used to generate seed alignments for *Alphaherpesvirinae* (HSV-1, VZV) or across all sub-families (HSV-1, VZV, HCMV, HHV7, KSHV, EBV). Seed alignments were used to generate hidden Markov models (HMMs) using the HMMER (*Eddy, 2011*) program hmmbuild. HMMs were subsequently used to extract and align homologue sequences from UniProt using HMMER (*Eddy, 2011*) program hmmsearch locally (for *Alphaherpesvirinae*) or using the HMMER web server (*Finn et al., 2011*) (for all *Herpesviridae*). We mapped the proteins thus identified to the source virus genomes, discarding any protein sequences from partial genome sequences where pUL7 or pUL51 were absent. Our initial alignments comprised distinct pairs of pUL7 and pUL51 sequences from 205 *Alphaherpesvirinae*, 147 *Betaherpesvirinae* and 78 *Gammaherpesvirinae*, and the alignments for homologues in each subfamily were improved by manual correction.

The structure of the core pUL7:pUL51 heterodimer was inspected to compile a table of 63 pairwise interactions between amino acids in the two proteins, 59 of which involved side chain atoms. These interactions arose from 33 distinct residues in pUL7 and 29 residues in pUL51. Using the alignments generated above, we compiled a matrix of amino acid pairs (one in each pUL7 and pUL51 homologue) that are predicted to interact. For each pair of interacting sites, we calculated the strength of the correlation between its amino acid states across the alignment. For this purpose, we followed Zaykin and colleagues (equation 3 of *Zaykin et al., 2008*). For a single pair of sites, whose alignments contain, respectively, $k$ and $m$ amino acid states, then the correlation between two of those states, $i$ and $j$, is

$$r_{ij} = \frac{p_{ij} - p_i p_j}{\sqrt{p_i(1-p_i)}\sqrt{p_j(1-p_j)}}$$

where $p_i$ is the proportion of strains that carry amino acid $i$ at the relevant site in pUL7, $p_j$ is the proportion that carry amino acid $j$ in pUL51, and $p_{ij}$ is the proportion of strains that carry both. The total strength of correlation at the pair, $T$, is

$$T = \frac{(k-1)(m-1)}{km} N \sum_{i=1}^{k} \sum_{j=1}^{m} r_{ij}^2$$

where $N$ is the number of strains, and the test statistic, $z$, is this quantity summed across all interacting pairs

$$z = \sum_{i=1}^{I} T$$

where $I$ is the number of interactions. To test whether $z$, the signature of coevolution, was significantly greater than would be expected by chance, we compared the measured test statistic to a null distribution comprised of $10^6$ data sets for which the interacting partner sites were randomly permuted. The $p$ value for each test was the proportion of randomly permuted data sets for which the test statistic was greater than or equal to the value for the real data. Under our permutation scheme, each randomized data set resembled the true data in terms of the total number of interactions, the number of interactions involving each site, and the allele frequencies at each putatively interacting site. The test also controls for shared evolutionary history, which can generate spurious evidence of coevolution (*Horner et al., 2008*). As a consequence, however, the test is expected to be highly conservative, because many of the randomized interactions might resemble the true interactions (not least because single sites were involved in multiple putative interactions) and because, under plausible evolutionary scenarios, multiple interacting pairs might evolve in concert.

Of this set of interactions, not all could be analyzed for all sequences, either because of missing amino acids in some sequences (due to both deletions and missing data), or because we could not be confident in the alignment of some sites. There was thus an inherent trade-off between maximizing the number of interactions and maximizing the number of strains in the test. We initially examined alignments across the *Herpesviridae*, but the low sequence identity meant that we could not confidently assign homology for most sites involved in putative interactions. Across the family as a whole, only 12 conserved interacting pairs could be analyzed, and this led to an underpowered test. Accordingly, we restricted our analyses to the *Alphaherpesvirinae*. From our initial alignments we excluded six very short sequences (one pUL51 homologue: A0A2Z4H851, and five pUL7 homologue: A0A120I2R6, A0A097HXP5, A0A286MM74, A0A2Z4H5E9, A0A120I2N0). This led to an alignment containing 199 strains, for which the amino acids of 35/63 interacting sites could be confidently aligned across all strains. These 35 interactions involved 21 sites from pUL51 and 19 sites from pUL7 (main text and *Data set 1*; *Supplementary file 1*–Table S4).

Because so many interactions were missing from this analysis, we next excluded two further pUL7 homologue sequences (B7FEJ7, A0A0X8E9M8) where many of the interacting sites could not be confidently aligned. This led to an alignment of 197 strains, for which 54/63 interactions could be tested (involving all 29 putatively interacting sites from pUL51 and 29/33 sites from pUL7). Despite the increase in the size of the data set, results were little changed (*Data set 2*; *Supplementary file 1*–Table S4). Results were similarly little changed when we considered only interactions involving side chain atoms (*Data set 3*; *Supplementary file 1*–Table S4), and when we restricted our analysis to the subset of better conserved positions, as found in the regions of aligned sequence returned by HMMER (*Data set 4*; *Supplementary file 1*–Table S4). *R* code for performing the analysis is available as file *Source code 1*. Sequence alignments and table of interacting residues are available in *Source data 1*.

## Mammalian cell culture and transfection

Mycoplasma-free human embryonic kidney (HEK) 293 T and HeLa cells were maintained in Dulbecco's modified Eagle's medium (DMEM; ThermoFisher) supplemented with 10% (v/v) heat-inactivated fetal bovine serum (FBS) and 2 mM L-glutamine (ThermoFisher). Cells were maintained at 37°C in a humidified 5% $CO_2$ atmosphere.

Plasmids for GFP-pUL7 (N-terminal tag) and pUL51-mCherry (C-terminal tag) were as used in *Albecka et al., 2017*. Homologues from other herpesviruses were cloned into pEGFP-C2, encoding an N-terminal GFP tag, or pmCherry-N1, encoding a C-terminal mCherry tag, as follows. pUL103 and pUL71 were cloned from HCMV strain Toledo cDNA, pORF42 and pORF55 were cloned from KSHV strain JSC-1 cDNA, and VZV pORF53 and pORF7 were cloned from codon-optimized synthetic genes (GeneArt) to boost their otherwise-poor expression in cultured cells.

For co-precipitation experiments, $5 \times 10^6$ HEK 293 T cells were transfected by adding 1 µg total DNA (split evenly by mass between the plasmids indicated) and 1.5 µg of branched polyethylenimine (PEI; average MW ~25,000, Merck) that had been diluted in Opti-MEM (ThermoFisher) and incubated together for 20 min before addition to cells.

For immunocytochemistry, $7.5 \times 10^4$ HeLa cells/well were seeded in six-well plates containing four sterile no. 1.5 coverslips/well and grown overnight before being transfected by addition of 625 ng total DNA (split evenly by mass between the plasmids indicated) and 6 μL/well TransIT-LT1 (Mirus) that had been diluted in Opti-MEM and incubated together for 20 min before addition to cells.

## Co-precipitation and immunoblotting

Cells were harvested 24 hr post-transfection by scraping in phosphate buffered saline (PBS; 137 mM NaCl, 2.7 mM KCl, 10 mM $Na_2HPO_4$, 1.8 mM $KH_2PO_4$), and washed twice in PBS. Cell pellets were resuspended in lysis buffer (10 mM tris pH 7.5, 150 mM NaCl, 0.5 mM EDTA, 0.5% IGEPAL CA-630 (a.k.a. NP-40, Merck), 1% (v/v) EDTA-free protease inhibitor cocktail (Merck)) and incubated at 4°C for 30 min before clarification by centrifugation at $20,000 \times g$, 4°C for 10 min. The protein concentration in each lysate was normalized after assessment using the bicinchoninic acid assay (ThermoFisher) according to the manufacturer's instructions. Normalised lysates were incubated for 1 hr at 4°C with GFP-Trap or RFP-Trap bead slurry (Chromotek) that had been pre-equilibrated in wash buffer (10 mM tris pH 7.5, 150 mM NaCl, 0.5 mM EDTA). Following incubation, beads were washed three times, the supernatant was completely removed, beads were resuspended in SDS-PAGE loading buffer and the samples were heated at 95°C for 5 min to liberate bound proteins before removal of the beads by centrifugation. Samples were separated by SDS-PAGE using 12% or 15% polyacrylamide gels and transferred to Protran nitrocellulose membranes (Perkin Elmer) using the Mini-PROTEAN and Mini-Trans-Blot systems (BioRad) following the manufacturer's protocol. After blocking in PBS with 5% (w/v) non-fat milk powder, membranes were incubated with primary antibody overnight at 4°C and then secondary antibody for 1 hr at room temperature. Dried blots were visualized on an Odyssey CLx infrared scanner (LI-COR).

## Immunocytochemistry

Cells were transferred onto ice 24 hr post-transfection. Coverslips were washed with ice-cold PBS and incubated with cold 250 mM HEPES pH 7.5, 4% (v/v) electron microscopy-grade formaldehyde (PFA, Polysciences) for 5 min on ice before being incubated with 250 mM HEPES pH 7.5, 8% (v/v) PFA at room temperature for 10 min. Coverslips were washed with PBS before quenching of residual PFA by addition of 25 mM $NH_4Cl$ for 5 min at room temperature. After washing with PBS, cells were permeabilized by incubation with 0.1% saponin in PBS for 30 min before being incubated with blocking buffer (5% (v/v) FBS, 0.1% saponin in PBS) for 30 min. Primary antibodies (below) were diluted in blocking buffer and incubated with coverslips for 2 hr. Coverslips were washed five times with blocking buffer before incubation for 1 hr with the relevant secondary antibodies (below) diluted in blocking buffer. Coverslips were washed five times with blocking buffer, three times with 0.1% saponin in PBS, three times with PBS, and finally with ultrapure water. Coverslips were mounted using Mowiol 4–88 (Merck) containing 200 nM 4′,6-diamidino-2-phenylindole (DAPI) and allowed to set overnight. Images were acquired using a Zeiss LSM780 confocal laser scanning microscopy system mounted on an AxioObserver.Z1 inverted microscope using a 64× Plan Apochromat objective (NA 1.4). Images were processed using Fiji (*Rueden et al., 2017*; *Schindelin et al., 2012*).

## Antibodies

Primary antibodies used for immunoblotting were rabbit anti-GFP (Merck, G1544), rat anti-RFP (Chromotek, 5F8), or mouse anti-GAPDH (ThermoFisher, AM4300). Secondary antibodies for immunoblotting were LI-COR IRDye 680T conjugated goat anti-rat (926–68029), donkey anti-rabbit (926–68023) or goat anti-mouse (926–68020), or LI-COR IRDye 800CW conjugated donkey anti-rabbit (926–32213) or goat anti-mouse (926-32210). Primary antibodies used for immunocytochemistry were anti-TGN46 (Bio-Rad, AHP500G), mouse anti-Paxillin (BD Biosciences, 610051), rabbit anti-Zyxin (abcam, ab71842), and secondary antibodies were Alexa Fluor 647 conjugated donkey anti-sheep (A-21448, ThermoFisher), goat anti-mouse (A-21236, ThermoFisher) or goat anti-rabbit (A-21245, ThermoFisher).

## Data availability

Crystallographic coordinates and structure factors have been deposited in the Protein Data Bank, www.pdb.org (accession code 6T5A), and raw diffraction images have been deposited in the University of Cambridge Apollo repository (https://doi.org/10.17863/CAM.44914). SAXS data, *ab initio* models and pseudo-atomic models have been deposited into the Small-Angle Scattering Biological Data Bank (SASBDB) (*Valentini et al., 2015*) under the accession codes SASDG37 (pUL7:pUL51(8–142) heterotrimer), SASDG47 (pUL7:pUL51 heterohexamer) and SASDG57 (pUL7:pUL51 heterotrimer). Mass spectrometry data have been deposited to the ProteomeXchange Consortium via the PRIDE (*Perez-Riverol et al., 2019*) partner repository with the dataset identifier PXD015941.

## Acknowledgements

HCMV and KSHV cDNA were kind gifts of John Sinclair and Mike Gill. We thank Janet Deane for access to MALS equipment, Janet Deane and the mentors at the DLS-CCP4 Data Collection and Structure Solution Workshop 2018 for helpful discussions, Len Packman for peptide fingerprinting mass spectroscopy analysis, Susanna Colaco, Heather Coleman and Viv Connor for superb technical assistance, Nick Bright for assistance with electron microscopy, Diamond Light Source for access to beamlines I03 and I04 under proposal mx15916, and EMBL for access to the bioSAXS beamline P12 under proposal HH-SAXS-911. Remote synchrotron access was supported in part by the EU FP7 infrastructure grant BIOSTRUCT-X (Contract No. 283570) and access to P12 was supported by iNEXT funded by the Horizon 2020 programme of the European Commission (grant number 653706). A Titan V graphics card used for this research was donated by the NVIDIA Corporation. BGB is a Wellcome Trust PhD student, DJO was supported by a John Lucas Walker Studentship, and MFA was supported by Commonwealth Scholarship Commission PhD scholarship (BDCA-2014–7). This work was supported by a Sir Henry Dale Fellowship (098406/Z/12/B), jointly funded by the Wellcome Trust and the Royal Society (to SCG).

## Additional information

### Funding

| Funder | Grant reference number | Author |
| --- | --- | --- |
| Wellcome | 098406/Z/12/B | Stephen C Graham |
| Royal Society | 098406/Z/12/B | Stephen C Graham |
| Nvidia | | Stephen C Graham |
| Wellcome Trust | | Benjamin G Butt |
| John Lucas Walker Studentship | | Danielle J Owen |
| Commonwealth Scholarship Commission | | Md Firoz Ahmed |

The funders had no role in study design, data collection and interpretation, or the decision to submit the work for publication.

### Author contributions

Benjamin G Butt, Data curation, Software, Validation, Investigation, Visualization, Writing - original draft, Writing - review and editing; Danielle J Owen, Lyudmila Ivanova, Jack W Houghton, Investigation, Writing - review and editing; Cy M Jeffries, Chris H Hill, Investigation, Methodology, Writing - review and editing; Md Firoz Ahmed, Validation, Investigation, Writing - review and editing; Robin Antrobus, Methodology, Writing - review and editing; Dmitri I Svergun, Funding acquisition, Methodology, Writing - review and editing; John J Welch, Software, Formal analysis, Methodology, Writing - review and editing; Colin M Crump, Conceptualization, Resources, Funding acquisition, Writing - review and editing; Stephen C Graham, Conceptualization, Resources, Data curation, Software, Supervision, Funding acquisition, Investigation, Visualization, Writing - original draft, Project administration, Writing - review and editing

## Author ORCIDs
Benjamin G Butt [iD] https://orcid.org/0000-0001-6718-0470
Chris H Hill [iD] http://orcid.org/0000-0001-7037-0611
Colin M Crump [iD] http://orcid.org/0000-0001-9918-9998
Stephen C Graham [iD] https://orcid.org/0000-0003-4547-4034

## Decision letter and Author response
Decision letter https://doi.org/10.7554/eLife.53789.sa1
Author response https://doi.org/10.7554/eLife.53789.sa2

# Additional files
## Supplementary files
• Source code 1. Code for performing evolutionary analysis of α-herpesvirus pUL7:pUL51 interaction interface. R code is in coevolution-test.R. To perform analyses on *Data sets 1–4* as reported in *Supplementary file 1*–Table S4 use *Source data 1* and set the variable 'mydataset' accordingly.

• Source data 1. Source data for evolutionary analysis of α-herpesvirus pUL7:pUL51 interaction interface. Zip file contains alignments of pUL7 and pUL51 homologue sequences from across *Alphaherpesvirinae* (ul7.alpha.alignment.fas and ul51.alpha.alignment.fas, respectively), the restricted pUL7 alignments across the subset of sequences returned by HMMER (ul7.alpha.HMMER.alignment.fas), the table of interactions between pUL7 and pUL51 residues (InteractionsLookup.txt), the table of per-species pUL7 and pUL51 sequences (virgroups.txt), and files to match the annotated interaction sites to the multiple alignment (UL7.alpha.Rdata, UL7.alpha.HMMER.Rdata and UL51.alpha.Rdata).

• Supplementary file 1. Data tables. Contains tables with data collection parameters for the SAXS and X-ray diffraction experiments, list of cross-linked peptides identified by mass spectrometry, and details of the pUL7-pUL51 co-evolution analysis.

• Transparent reporting form

## Data availability
Crystallographic coordinates and structure factors have been deposited in the Protein Data Bank, www.pdb.org (accession code 6T5A), and raw diffraction images have been deposited in the University of Cambridge Apollo repository (https://doi.org/10.17863/CAM.44914). SAXS data, ab initio models and pseudo-atomic models have been deposited into the Small-Angle Scattering Biological Data Bank (SASBDB) under the accession codes SASDG37 (pUL7:pUL51(8-142) heterotrimer; https://www.sasbdb.org/data/SASDG37/), SASDG47 (pUL7:pUL51 heterohexamer; https://www.sasbdb.org/data/SASDG47/) and SASDG57 (pUL7:pUL51 heterotrimer; https://www.sasbdb.org/data/SASDG57/). Mass spectrometry data have been deposited to the ProteomeXchange Consortium via the PRIDE partner repository with the dataset identifier PXD015941. Source data and code for performing evolutionary analysis of the pUL7:pUL51 interaction interface across α-herpesviruses is provided in files Source code 1 and Source data 1.

The following datasets were generated:

| Author(s) | Year | Dataset title | Dataset URL | Database and Identifier |
|---|---|---|---|---|
| Butt BG, Jeffries CM, Graham SC | 2020 | 2:4 heterohexamer of pUL7 and pUL51 from herpes simplex virus 1 | https://www.sasbdb.org/data/SASDG47/ | SASBDB, *SASDG47* |
| Butt BG, Jeffries CM, Graham SC | 2020 | 1:2 heterotrimer of pUL7 and pUL51(8-142) from herpes simplex virus 1 | https://www.sasbdb.org/data/SASDG37/ | SASBDB, *SASDG37* |
| Butt BG, Graham SC | 2020 | Crystal structure of herpes simplex virus 1 pUL7:pUL51 complex | https://www.rcsb.org/structure/6T5A | RCSB Protein Data Bank, 6T5A |
| Butt BG, Graham SC | 2019 | Crystallographic diffraction data for the structure of herpes simplex | https://doi.org/10.17863/CAM.44914 | University of Cambridge Apollo |

| | | | | |
|---|---|---|---|---|
| | | virus 1 pUL7:pUL51 complex | | repository, 10.17863/CAM.44914 |
| Graham SC, Houghton JW | 2020 | Structure of herpes simplex virus pUL7:pUL51, a conserved complex required for efficient herpesvirus assembly | http://proteomecentral.proteomexchange.org/cgi/GetDataset?ID=PXD015941 | ProteomeXchange, PXD015941 |
| Butt BG, Jeffries CM, Graham SC | 2020 | 1:2 heterotrimer of pUL7 and pUL51 from herpes simplex virus 1 | https://www.sasbdb.org/data/SASDG57/ | SASBDB, *SASDG57* |

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
