## [Decision Letter]

**Acceptance summary:**

Graham and colleagues describe the structural characterization of the Herpes virus tegument protein complex pUL7-pUL51, which is important for membrane envelopment of the capsid in the cytosol and is conserved across herpesviruses. The most exciting observation that truncated pUL51 is structurally similar to the helical hairpin formed by CHMP4B, the most abundant subunit of the ESCRT-III membrane fission complex. The authors therefore propose that pUL51 may function in viral envelopment and budding by mimicking (or recruiting) the cellular ESCRT-III complex, which is used by many other enveloped viruses to bud from cells. Although this hypothesis remains to be tested functionally, the authors provide strong support by demonstrated that truncated pUL51 proteins can form helical filaments that resemble those formed by ESCRT-III proteins.

**Decision letter after peer review:**

Thank you for submitting your article "Insights into herpesvirus assembly from the structure of the pUL7:pUL51 complex" for consideration by *eLife*. Your article has been reviewed by two peer reviewers, and the evaluation has been overseen by a Wes Sundquist, Reviewing Editor and Cynthia Wolberger, Senior Editor. The reviewers have opted to remain anonymous.

The reviewers have discussed the reviews with one another and the Reviewing Editor has drafted this decision to help you prepare a revised submission.

Summary:

The manuscript by Graham and colleagues describes the structural characterization of the Herpes virus tegument protein complex pUL7-pUL51, which is important for membrane envelopment of the capsid in the cytosol. The complex is conserved across herpesviruses.

The major findings are:

1) SAXS analysis indicates that pUL7:pUL51 forms a 1:2 complex with an elongated shape likely due to an unstructured C-terminal region of pUL51.

2) Deletion of the pUL51 C-terminus resulted still in a 2:1 pUL7-pUL51(1-170) complex, which was characterized by SAXS and shown to be more compact.

3) An even smaller pUL7-pUL51(8-142) complex forms a 1:1 complex and its structure was determined at high resolution by X-ray crystallography.

4) The pUL7 structure adopts a novel, predominantly helical fold that the authors term a "Conserved UL7(Seven) Tegument Assembly/Release Domain (CUSTARD)" fold

5) The structure of the truncated pUL51 construct reveals a surprising feature that could provide an essential clue for understanding its critical function in viral replication because there is striking similarity between pUL51 and the helical hairpin formed by CHMP4B, a subunit of the ESCRT-III membrane fission complex. The authors therefore propose that pUL51 may function in viral envelopment and budding by mimicking (or recruiting) the cellular ESCRT-III complex.

5) Based on the heterodimer data, chemical cross linking, Mass spec and the SAXS data a model of the pUL7-pUL51 complex was reconstructed to position the missing C-terminal region of pUL51.

6) Part of the interface of the pUL7-pUL51 complex is conserved across herpesviruses, and bioinformatics analysis suggest co-evolution of interacting residue pairs.

7) pUL51 does not interact with pUL14 as had been suggested previously.

8) Co-localization of pUL7-pUL51 with focal adhesion markers paxillin and zyxin as previously suggested, was not observed.

Essential revisions:

Overall, this is likely to be an important piece of work for understanding the mechanism of envelopment of Herpesviruses. The manuscript is well written, the data are of high technical quality, and the structure is novel and presents a new fold of pUL7. However, the study currently lacks experimentally-based insights into how the pUL7-pUL51 complex functions in membrane envelopment. In particular, the hypothesis that pUL51 functions by mimicking CHMP4B needs to be supported by functional evidence because this is the most important claim in the paper. Given the strong similarity between pUL51 and CHMP4B, the modeling of key pUL51 residues predicted to function in membrane binding and remodeling should be straightforward.

To substantiate the ESCRT-III hypothesis, the authors need to provide convincing functional data on pUL51. Relevent experiments might include testing membrane binding and polymerization in vitro, testing Golgi membrane binding in cells using structure-based mutagenesis, and evaluating relevant mutations in the context of viral replication. The key questions that need to be answered are:

1) Does pUL51 bind to membranes without palmitoylation and, more importantly, is the binding mode consistent with the ESCRT-III model? It is not obvious from the figures that the complex can be positioned on a membrane via the helical hairpin in the complex.

2) Does pUL51 form ESCRT-III-like filaments. The authors report that pUL51 forms large aggregates in vitro. Are these aggregates ESCRT-III-like filaments?

3) How does pUL7 affect pUL51 polymerization and membrane binding?

If the authors are able to provide these types of convincing functional studies, then the manuscript will become a conclusive work that advances the field significantly and should be acceptable for publication in *eLife*.

---

## [Author Response]

Essential revisions:Overall, this is likely to be an important piece of work for understanding the mechanism of envelopment of Herpesviruses. The manuscript is well written, the data are of high technical quality, and the structure is novel and presents a new fold of pUL7. However, the study currently lacks experimentally-based insights into how the pUL7-pUL51 complex functions in membrane envelopment. In particular, the hypothesis that pUL51 functions by mimicking CHMP4B needs to be supported by functional evidence because this is the most important claim in the paper. Given the strong similarity between pUL51 and CHMP4B, the modeling of key pUL51 residues predicted to function in membrane binding and remodeling should be straightforward.To substantiate the ESCRT-III hypothesis, the authors need to provide convincing functional data on pUL51. Relevent experiments might include testing membrane binding and polymerization in vitro, testing Golgi membrane binding in cells using structure-based mutagenesis, and evaluating relevant mutations in the context of viral replication. The key questions that need to be answered are:1) Does pUL51 bind to membranes without palmitoylation and, more importantly, is the binding mode consistent with the ESCRT-III model? It is not obvious from the figures that the complex can be positioned on a membrane via the helical hairpin in the complex.

It has been published previously that pUL51 does not readily associate with membranes when its palmitoylation is blocked by mutation of the acceptor cysteine residue to serine or alanine [Nozawa et al., 2003]. The palmitoylation site of pUL51 (residue 9) is located well before the first residue of pUL51 in the pUL7:pUL51 core heterodimeric complex (residue 41, Figure 2B). While it is likely that helix 1 of pUL51 is more extended in the polymeric form of the protein (consistent with the secondary structure predictions in Figure 1—figure supplement 2 and the new circular dichroism data presented in Figure 5—figure supplement 1), there would still be approximately 16 amino acids between this palmitoylated Cys residue and the predicted start of the core ESCRT-III–like helical hairpin. We posit that the N-terminal palmitoylation of pUL51 and homologues is akin to the ANCHR motif of Snf7 and the myristylation of the CHMP4B nucleating protein CHMP6, and that pUL51 therefore fulfils the functions of both CHMP6 and CHMP4B in infected cells. This hypothesis is consistent with observations that ‘canonical’ ESCRT-III initiation signals, mediated by ESCRT-I, ESCRT-II or the Bro1-domain containing proteins, are dispensable for HSV-1 membrane envelopment [Barnes and Wilson, 2020 and Pawliczek and Crump, 2009].

We have included a schematic diagram in Figure 5 and have updated the Discussion to clarify our proposal for pUL51 polymerisation and its role in virus assembly. We have also clarified our proposal that pUL7 is a negative regul,ator of pUL51 polymerisation, blocking the pUL51 self-association interface akin to the regulation of Shrub polymerisation by Lgd [McMillan et al., 2017] or CHMP4B by CC2D1A [Martinelli et al., 2012].

2) Does pUL51 form ESCRT-III-like filaments. The authors report that pUL51 forms large aggregates in vitro. Are these aggregates ESCRT-III-like filaments?

We have now shown that a truncated form of pUL51, including the N-terminal regions predicted to be helical (residues 1-170), forms ESCRT-III like filaments. We initially concentrated our efforts on assaying the polymerisation of the full-length pUL51 protein shown in Figure 1—figure supplement 1. However, the aggregation-prone nature of this protein made it exceedingly challenging to work with – the yields were very low and it would irreversibly aggregate before we could apply it to EM grids. We therefore designed a new purification strategy by refolding pUL51(1-170) from *E. coli* inclusion bodies. As shown in Figure 5—figure supplement 1, this refolded protein is pure and largely helical, and as shown in Figure 5 this protein forms ESCRT-III like filaments.

3) How does pUL7 affect pUL51 polymerization and membrane binding?

Our previously-published work shows that pUL51 recruits pUL7 to membranes (Figure 4 of Albecka et al., 2017). Based on the similar orientations of Lgd DM14-3 relative to Shrub and pUL7 relative to pUL51, we predict that pUL7 will inhibit pUL51 polymerisation. We planned to test this hypothesis using a ‘poisoning’ method, by adding increasing concentrations of pUL7:pUL51(8-142) to pure pUL51(1-170) and monitoring the effect upon pUL51(1-170) polymerisation in vitro. However, given the challenges in producing sufficient pure pUL51, these polymerisation assays were optimised only days before our laboratory was closed due to the COVID-19 outbreak. We were therefore unable to perform these poisoning experiments. We hope to perform these experiments as soon as we are back in the lab, in addition to probing pUL51 mutants that were designed to block polymerisation based on the structural homology to cellular ESCRT-III components. We humbly suggest that these experiments may be suitable for publication as a Research Advance in *eLife* at a later date.